# Direct visualization of solute locations in laboratory ice samples

Ted Hullar[1], Cort Anastasio[1]

[1] Department of Land, Air, and Water Resources, University of California, Davis

Correspondence to Cort Anastasio (canastasio@ucdavis.edu)

**Abstract**

Many important chemical reactions occur in polar snow, where solutes may be present in several reservoirs, including at the air-ice interface and in liquid-like regions within the ice matrix. Some recent laboratory studies suggest chemical reaction rates may differ in these two reservoirs.  While investigations have examined where solutes are found in natural snow and ice, few studies have examined solute locations in laboratory samples, nor the possible factors controlling solute segregation. To address this, we used micro-computed tomography (microCT) to examine solute locations in ice samples prepared from either aqueous cesium chloride (CsCl) or Rose Bengal solutions that were frozen using several different methods.  Samples frozen in a laboratory freezer had the largest liquid-like inclusions and air bubbles, while samples frozen in a custom freeze chamber had somewhat smaller air bubbles and inclusions; in contrast, samples frozen in liquid nitrogen showed much smaller concentrated inclusions and air bubbles, only slightly larger than the resolution limit of our images (~2 µm).  Freezing solutions in plastic versus glass vials had significant impacts on the sample structure, perhaps because the poor heat conductivity of plastic vials changes how heat is removed from the sample as it cools.  Similarly, the choice of solute had a significant impact on sample structure, with Rose Bengal solutions yielding smaller inclusions and air bubbles compared to CsCl solutions frozen using the same method.  Additional experiments using higher-resolution imaging of an ice sample show that CsCl moves in a thermal gradient, supporting the idea that the solutes in ice are present in mobile liquid-like regions. Our work shows that the structure of laboratory ice samples, including the location of solutes, is sensitive to freezing method, sample container, and solute characteristics, requiring careful experimental design and interpretation of results.

## 1.  Introduction

Snowpacks can be important locations for a variety of chemical reactions, particularly in polar regions (Bartels-Rausch et al., 2014; Domine and Shepson, 2002).  Because light can penetrate several 10s of cm into the snowpack, photochemical reactions are particularly important (Grannas et al., 2007), including nitrate photolysis forming $NO_x$ (Beine et al., 2002; Chu and Anastasio, 2003; Jacobi et al., 2004), hydrogen peroxide photolysis forming hydroxyl radical (Chu and Anastasio, 2005; Jacobi et al., 2006), and transformation of organics (Dibb and Arsenault, 2002; Sumner and Shepson, 1999).

A variety of potential chemical reactants have been identified in snowpacks; concentrations can vary considerably, with typical concentrations on the order of 10 µM in clean Arctic snows (Yang et al., 1996).  Impurities can integrate into snow crystals during formation, or be deposited onto the surface of formed crystals.  Reactants and products also partition between the snow crystals and the overlying air; the large surface area of the snow crystals provides an extensive environment for reactions to occur.  As the snowpack consolidates and snow grains metamorphose, chemical compounds can remain at the

surface of the crystals, or become trapped internally at grain boundaries or triple junctions (Bartels-
Rausch et al., 2014; Domine et al., 2008; Grannas et al., 2007).
There appear to be three reservoirs for impurities in snow:  a quasi-liquid layer (QLL) at the ice-
air interface; liquid-like regions (LLRs) within the ice (e.g., at grain boundaries); and in  the bulk ice
matrix, i.e., between frozen water molecules  (Barret et al., 2011; Grannas et al., 2007; Jacobi et al.,
2004).  While the exact location of solutes in snow is not well understood (Bartels-Rausch et al., 2014),
the location is important for several reasons.  First, chemicals in a surface QLL can be more readily
released to the atmosphere compared to impurities segregated into an internal LLR; furthermore, gas-
phase oxidants and other species can readily partition from the air onto solutes at the air-ice interface.
Second, photon fluxes can vary considerably in various locations within the snowpack (Phillips and
Simpson, 2005), although there appear to be only small differences within crystals themselves (McFall
and Anastasio, 2016).  Third, the rates of reactions of impurities appear to vary with location.  For
example, photolysis rates of PAHs (polycyclic aromatic hydrocarbons) have been reported to be up to five
times faster in surface QLLs compared to in whole ice samples (where PAHs are likely in LLRs) or in
aqueous solution (Kahan and Donaldson, 2007, 2010; Ram and Anastasio, 2009).  An investigation of
reactions in frozen solutions (Kurkova et al., 2011) suggested the QLL and LLR physical reaction
environments are substantially different, with QLLs best represented by a 2D cage and LLRs as a 3D
cage.  This work also found that the cage effect (i.e., the tendency for a compound to be surrounded by
solvent molecules, which can impede the ability of a compound to react) at a given temperature was much
more pronounced for reactions occurring in QLLs than LLRs, with solutes in QLLs having less mobility
compared to solutes in LLRs.
Because of the potential reactivity differences between the reservoirs, understanding reaction
rates in different reservoirs requires knowing where solutes are located.  Solute locations in natural snow
and ice samples have been studied using electron microscopy (Barnes et al., 2003; Lomonaco et al., 2009;
Rosenthal et al., 2007), and were found to preferentially segregate to grain boundaries and triple
junctions.  Additional work has evaluated the nature of these compartments, showing that solutes
segregate and concentrate in LLRs (Heger et al., 2005; Heger et al., 2006).  When an aqueous solution is
frozen, most solutes are excluded from the forming ice matrix (Hobbs, 1974; Petrenko and Whitworth,
1999), often forming platelets of ice separated by brine or dendritic structures (Rohatgi and Adams, 1967;
Shumskii, 1964). Recently, some studies have used various techniques to directly examine the location of
solutes themselves in laboratory snow and ice samples (Cheng et al., 2010; Miedaner, 2007; Miedaner et
al., 2007) Nonetheless, solute location is poorly understood in many experimental systems, and , is most
often inferred from the way the sample is made (Kahan et al., 2010) or from chemical behavior (Kurkova
et al., 2011).
The main goal of this paper is to examine the location of solutes in laboratory-prepared frozen
solutions.  In order to do this, we use X-ray computed tomography (CT), a technique that has been used to
create 3-dimensional images of a variety of biological and natural materials (Blanke et al., 2013; Evans et
al., 2008).  High resolution microCT, which is capable of a spatial resolution of < 10 μm, has been used to
look at the structure of natural snow and ice (Chen and Baker, 2010; Heggli et al., 2011; Lomonaco et al.,
2011; Obbard et al., 2009).  But to our knowledge this method has not been used to investigate the
structure and solute locations for laboratory samples prepared under reproducible conditions with specific
solutes.
Thus here we examine the locations of impurities in frozen aqueous solutions prepared in the
laboratory.  We are primarily interested in the locations of solutes in ices prepared using different freezing
methods aimed at putting solutes in specific reservoirs within the ice; these methods, or similar ones, have
been used both in our previous research as well as by other investigators.  In this work we focus on
cesium chloride (CsCl) as our solute.  However, because previous studies (Cheng et al., 2010; Rohatgi
and Adams, 1967) found different solutes can affect freezing morphology and therefore may influence
solute location, we also imaged ice containing the organic compound Rose Bengal (4,5,6,7-tetrachloro-
2',4',5',7'-tetraiodofluorescein).  For our samples we present both qualitative (visual) and semi-
quantitative (tabular and graphical) results.
## 2.  Methods
We prepared samples by freezing 1.0 mM aqueous solutions of cesium chloride or, in a few
cases, 1.0 mM Rose Bengal.  High purity water ("Milli-Q water") was produced from house-treated
deionized water that was run through a Barnstead International DO813 activated carbon cartridge and
then a Millipore Milli-Q Plus system.  We chose cesium chloride (Sigma-Aldrich, 99.9%) for our primary
solute because of its high solubility in water and high X-ray mass attenuation coefficient ($\sim$4.4 $cm^2$ $g^{-1}$ at
70 keV (NIST, 2015)), enabling visualization of low concentrations in our microCT system.  We also
used Rose Bengal to study the impacts of solute size and polarity on sample morphology.  While 1.0 mM
of solute is higher than typical total solute concentrations in continental (inland) natural snows, it is
within the range of concentrations measured in coastal snowpacks (Beine et al., 2011; Douglas and Sturm,
2004; Yang et al., 1996).   The chosen concentration allows easy visualization in our system and provides
enough material to evaluate spatial patterns in the sample.
We froze most samples as a 500 µl aliquot in a capped glass vial (approximately 3 cm high and 1
cm in diameter, 0.8 mm wall thickness, with a total vial volume of $\sim$2 ml) using one of three methods.
These methods were chosen because they had been used in our laboratory, as well as others, and also due
to differences in the speed of heat removal from the samples; we discuss later the expected morphologies
for the various sample types.  In the first technique ("Freezer"), we placed samples upright on a plastic
plate in a laboratory freezer at approximately -20° C; freezing took approximately 1 hour.  In the second
technique ("Freeze Chamber"), we froze samples upright in a custom-built freeze chamber (Hullar and
Anastasio, 2011) whose base was cooled to either -10 or -20° C.  Typically, the sample sat directly on the
base of the freeze chamber surrounded by air.  However, we also froze some samples surrounded by
drilled metal plates, effectively placing the sample in a metal "well"; the distance between the sample and
the surrounding plates was around 1 mm. In the third technique ("Liquid Nitrogen" or "LN2") we froze
samples by putting the aqueous sample in a vial, capping it, then immersing it in a bath of liquid nitrogen
deeper than the height of the liquid in the vial; freezing time was $\sim$30 seconds.  We allowed all samples to
anneal at -10° C for at least 1 hour before imaging. We froze a small number of samples in either
polypropylene vials (wall thickness $\sim$1 mm) or with a larger sample volume (750 µl).
We imaged samples using a MicroXCT-200 (Zeiss Instruments) micro-computed tomography
(microCT) scanner.  To maintain our samples at -10° C , samples were held in a custom cold stage for the
MicroXCT-200 (Hullar et al., 2014).  The custom cold stage was placed on the scanner's sample stage,
whose position is controlled by the scanner software to submicron precision.  Scanning parameters were
set based on the manufacturer's guidelines.  For most imaging, we set source and detector distances to 40
and 130 mm respectively; voltage and power were set at 70 keV and 7.9 W, and the manufacturer's LE3
custom filter was used for beam filtration.  The microCT acquired 1600 projections over 360 degrees of
rotation, with an exposure time of 2 s.  Images were reconstructed using the manufacturer's software on
an isotropic voxel grid with 15.9358 µm edge lengths.  Some samples were analyzed at higher resolution,
with a voxel edge length of 2.1146 μm. For these samples, we set source and detector distances to 60 and
18 mm, used the LE5 beam filter, collected 2400 projections spanning 360 degrees, and set beam voltage,
power, and exposure time to 60 keV, 6 W, and 30 s respectively. The microCT scanner software outputs
slicewise TIFF images of the X-Y plane of the sample, with greyscale values corresponding to the
radiodensity of each voxel at that Z plane.
We imported digital TIFF images into the Amira software package (Visualization Sciences
Group, FEI) for reconstruction and segmentation. Our segmentation procedure used the Amira
segmentation tools to isolate the sample from surrounding materials; generally, our procedure should
include very little sample container at the expense of excluding some small amounts of sample in contact
with the vial wall. Similarly, the segmentation procedure excludes very little sample in contact with air
above the sample, while including small amounts of top air as sample. Some images presented here were
mathematically smoothed by the software, which sometimes resulted in small features (< 80 μm in
diameter) being eliminated from movies and still images; however, smoothing did not substantially
change the interpretation of our results. In some cases we prepared histograms of the data, which were not
smoothed and include all sample data.
To quantitate CsCl concentration in each voxel, we first imaged samples of Milli-Q water, as both
liquid and ice, and measured the average radiodensity (image greyscale value) of a subvolume within
each sample. As expected, the average radiodensity of ice (4948 ± 160 (1σ)) was less than that of liquid
water (5372 ± 194 (1σ)) due to the lower density of ice. Our measured radiodensity ratio between ice (at -
10° C) and water (at 20° C) was 0.921, matching a calculated density ratio from literature values (Haynes,
2014) of 0.921. Next, we imaged 8 aqueous solutions of CsCl at varying concentrations (1.0 mM to 5.0
M) to construct a calibration curve. Plotting these points (Fig. 1) shows a linear relationship between
CsCl concentration and measured radiodensity, with a y-intercept value within the range of our measured
radiodensities for pure liquid water. Therefore, the measured radiodensity of a voxel within a sample
containing CsCl in solution (or ice) is linearly related to the amount of CsCl present in the voxel. We
assume the relationship between CsCl concentration and radiodensity is the same for ice and water. This
allows us to determine the amount of CsCl present in a sample voxel by subtracting the average greyscale
value of pure water (or ice) and then using the standard curve to calculate the CsCl mass.
When aqueous solutions are frozen, solutes are generally excluded from the forming ice matrix,
resulting in a two distinct components: pure (or nearly pure) water ice, and a concentrated solution of
solute (Cho et al., 2002; Lake and Lewis, 1970; Wettlaufer et al., 1997), which can be present at the air-
ice interface (i.e., as a QLL) and/or in LLRs within the sample. Freezing-point depression dictates that
the solute concentration in these regions is solely a function of the ice temperature (Cho et al., 2002) and
is independent of the solute concentration in the initial solution. For example, at -10° C, the predicted
total solute concentration in LLRs is 5.4 M of solute ions, or 2.7 M of a binary salt such as CsCl. This
LLR concentration is considerably lower than the solubility limit of CsCl (11.1 M at 20° C, 9.6 M at 0° C
(NIH, 2015)), but higher than the solubility limit of Rose Bengal (1 mM, temperature not given, (Neckers,
1989)). Therefore, we do not expect CsCl to precipitate, although Rose Bengal might.
As described earlier, we use the Fig. 1 calibration curve to convert microCT greyscale values of
radiodensity for each voxel to the mass of solute in each voxel. While this mass could be expressed as an
equivalent concentration in the voxel, we believe it is more accurate to consider each voxel as a mixture
of pure water ice (with zero solute) and LLRs (regions with a total solute ion concentration of 5.4 M at –
10° C, equivalent to a CsCl concentration of 2.7 M). Thus we express the composition of each voxel as
the fraction of voxel volume occupied by liquid-like regions, $V_{LLR}/V_{VOXEL}$:

$$\frac{V_{LLR}}{V_{VOXEL}} = \frac{(RD_{MEAS} - RD_{ICE})/_{Slope}}{2.7\ M} \tag{1}$$


where $V_{LLR}$ is the LLR volume, $V_{VOXEL}$ represents the volume of the entire voxel, $RD_{MEAS}$ is the

measured radiodensity of the voxel, $RD_{ICE}$ is the radiodensity of pure ice (4948), and Slope is the

measured slope of the standard curve line (10409 $M^{-1}$; Fig. 1). A voxel containing only pure ice has

$V_{LLR}/V_{VOXEL} = 0$, while a voxel composed entirely of 5.4 M total solute in water has $V_{LLR}/V_{VOXEL} = 1$.

Our estimated concentration of total solute ion concentration in LLRs is based on theoretical calculations

and assumes ideal behavior from the solution (Cho et al., 2002; Pruppacher and Klett, 2010). However, at

higher concentrations, solutions can deviate from ideal behavior. Pruppacher and Klett (Pruppacher and

Klett, 2010) and Haynes (Haynes, 2014) both present data for the freezing point depression of CsCl, but

only up to a salt concentration of 1.8 M (Pruppacher and Klett) or 1.4 M (Haynes). Extrapolating their

data to the concentrations expected in our samples (i.e., at -10 °C) suggest the CsCl concentration in

LLRs would be somewhere between 3 and 3.2 M, i.e., 10 – 20% higher than our ideal case concentration,

but neither source presents freezing point depression data measured at such a high concentration. In the

absence of measured information for the actual composition of CsCl solutions under our experimental

conditions, we have elected to stay with the theoretical prediction of salt concentration of 2.7 M. If the

actual LLR solute concentration is higher (lower) than 2.7 M, the $V_{LLR}/V_{VOXEL}$ values presented here

would be lower (higher); we estimate the largest magnitude of this error as approximately 20%. For

clarity, we use the measured $V_{LLR}/V_{VOXEL}$ values to segment many of our images into four domains:

voxels containing only air (defined as $V_{LLR}/V_{VOXEL} < -3.4\%$), voxels containing ice and little or no solute

($V_{LLR}/V_{VOXEL} = -3.4\%$ to 2%), voxels containing a moderate amount of solute ($V_{LLR}/V_{VOXEL} = $ 2-10%),

and voxels containing a substantial amount of solute ($V_{LLR}/V_{VOXEL} > 10\%$). We define an "air" voxel as

having a radiodensity less than or equal to the average radiodensity of an imaged air sample, i.e., 3996.

As noted above, greyscale values from images of pure materials vary somewhat, meaning a clear

distinction between two materials with similar average greyscale values is not possible. We chose to set

the cutoff for segmenting LLRs at a greyscale value of 5507, a threshold three standard deviations greater

than the average greyscale value for pure ice, which will essentially eliminate the problem of identifying

water ice as solute. However, because of this high threshold it is quite likely that solute is present in some

voxels characterized as "ice". On the other hand, voxels defined as having an LLR percentage of 2% or

greater almost certainly contain solute. For CsCl-containing samples, we calculated the mass of CsCl

present in each domain. Because the statistical distributions of voxels containing only pure water ice and

those containing <2% LLR as well as pure water ice overlapped , we could not determine the mass of

CsCl present in the "ice" domain directly. Therefore, we assumed any mass not present in either the LLR

2-10% or LLR >10% domains is present in the "ice" domain.

**3. Results and Discussion**

We first present imaging results for samples prepared without added solute (frozen Milli-Q

water). Figure 2a shows a reconstructed image of a "pure" ice sample prepared by freezing air-saturated

Milli-Q in a glass vial in a laboratory freezer; the full movie, which shows the sample rotating, is in

Supplemental Fig. S1. Air bubbles are visible as light grey spheroids, and are generally located towards

the center of the sample, away from the vial walls and base. This is likely because the entire outer surface

of the vial was cooled and the water apparently froze from the outside inward.  Supporting this idea, some
of the bubbles appear to elongate along the radial axes of the sample, similar to the bubble elongation
seen by Carte (Carte, 1961) in a temperature gradient.  The isolation of bubbles within the middle of the
sample seems to follow Shumskii's (Shumskii, 1964) model of the formation of the "central nucleus",
with impurities (in this case, air bubbles) forced to the center of a freezing water mass.
Figure 2b shows a reconstruction of a similar Milli-Q sample, but now where the solution was
degassed with helium for 30 minutes before freezing; the full movie is in Supplemental Fig. S2.  Because
He degassing replaces the more soluble nitrogen and oxygen in the air-saturated solution with less soluble
helium, fewer bubbles are present in Fig. 2b.  The size of the bubbles, however, is roughly similar in the
two figures (approximately 150-300 µm), suggesting bubble size is a function of the freezing method, not
of the gas itself.
Figure 2c shows a histogram of the number of voxels containing various radiodensities,
represented here as the ratio $V_{LLR}/V_{VOXEL}$, in the two water ice samples.   A ratio of zero represents the
average radiometric density for pure water ice, with values slightly greater or less than zero indicating
noise in the sample images and reconstruction.  Voxels containing only air comprise the smaller, second
peak centered at approximately $V_{LLR}/V_{VOXEL} = -0.05$, which overlaps with the primary (pure ice) peak.
Taking into account that the Y axis (voxel count) is a log scale, the two curves show the volume of gas
bubbles is clearly less for the helium degassed treatment.  Table 1 shows the estimated volumes of water
ice and gas bubbles in the two samples, as determined by our segmentation process (see the Methods
section).  The gas volume in ice made from air-saturated water is approximately 1.4 %, while the ice
made from helium-saturated Milli-Q has approximately half the gas volume.  Figures 2a and 2b appear to
show a larger difference in gas volume between the two samples, suggesting that many of the small
bubbles in the sample imaged in Fig. 2b may have been smoothed away and thus not visible.  For a
solution in equilibrium with air at 25 C, the mole fraction solubility of air (assuming a composition of
20% oxygen and 80% nitrogen) is $1.4 \times 10^{-5}$, while the value for helium is $7.0 \times 10^{-6}$ (Haynes, 2014), i.e.,
half the concentration of air in the solution.  The expected volume of bubbles in the helium degassed
treatment agrees well with the observed volume.
Next, we examined the effect of freezing method on both freezing morphology and solute
location.  The Freezer, Freeze Chamber, and LN2 sample preparation methods are described in the
Methods section.  Figure 3 shows the results of imaging several combinations of freezing method and
solute.  We start with an image of the ice made by freezing 1.0 mM CsCl in a laboratory freezer.  As
shown in Fig. 3a (and the Supplemental Fig. S3 movie), both air bubbles and concentrated CsCl LLRs are
relatively large, with the LLRs tending to wrap around the air bubbles.  Figure 3b is a magnification of
the red-bordered area in Fig. 3a, showing examples of large solute inclusions wrapped around air bubbles
(lighter gray spheroids).
Figure 3c (movie: Supplemental Fig. S4) shows a similarly prepared sample as the Freezer
sample in Fig. 3a, but frozen in our Freeze Chamber.  Compared to the Freezer sample, the Freeze
Chamber sample has smaller air bubbles and inclusions, more solute present near the top of the sample,
and the areas of concentrated solutes (LLRs) are less likely to be associated with the air bubbles.  These
points are clearly shown in Fig. 3d, which is a magnification of the red bordered area of Fig. 3c.
Considering that these two samples were frozen at similar temperatures, the morphologies are
substantially different.  As seen in Table 1, the fraction of voxels containing a LLR fraction >10% is
about five-fold less in the Freeze Chamber sample than the Freezer sample, while the fraction of voxels
with an LLR concentration between 2 and 10% doubles.  This finding indicates the freezing process in the
freeze chamber creates smaller LLR inclusions than does the freezer, with LLRs distributed more widely
throughout the sample.  Additionally, substantial amounts of solute were segregated towards the surface
of the Freeze Chamber sample; presumably, the sample froze from the bottom and solutes were
preferentially excluded from the advancing freezing front.  However, the same process did not affect the
air bubbles, which are well distributed throughout the sample. We believe these structural differences may
be due to faster freezing in the Freeze Chamber sample, as the freeze chamber removes heat more quickly
than the freezer because of direct contact between the bottom of the vial and the chilled base plate in the
chamber.  Previous work (Hallett, 1964; Rohatgi and Adams, 1967) has shown faster freezing gives closer
spacing of ice dendrites or plates in the sample as it freezes, which then leads to smaller solute inclusions
or bubbles, similar to our finding here.  Supplemental Fig. S5 shows a sample prepared in the same way
as in Fig. 3c, but with the metal plates in place in the freeze chamber, which surrounds the vial with metal
rather than air.  Here, we see similar bubble size and location as the sample frozen in the freeze chamber
without the metal plates.  However, unlike the sample frozen without plates in the freeze chamber, the
solute distribution with plates shows no segregation towards the top of the sample, probably because the
close proximity of the conductive metal plates removed heat from the sides and bottom of the sample
simultaneously, similar to the Freezer case.
Results for a 1.0 mM CsCl sample prepared with the third freezing method – liquid nitrogen – are
shown in Fig. 3e, with the full movie in Supplemental Fig. S6.  No air bubbles or significant solute
inclusions are visible.  However, as discussed earlier, some very small inclusions and air bubbles can be
removed by the mathematical smoothing done by the reconstruction software, so very small features (<
~80 μm) may be present in the sample but lost in the reconstruction.  A histogram of raw (i.e., not
smoothed) greyscale values from the LN2 sample image does show some voxels contain concentrated
solutes (Fig. 3g), as indicated by $V_{LLR}/V_{VOXEL}$ for some voxels towards the right-hand side of the graph
being greater than that of pure water ice.  As a further test of the possibility of solute inclusions in LN2
samples, we examined unreconstructed cross-sections of a 1.0 mM CsCl sample frozen in liquid nitrogen
and imaged at ~2 μm voxel resolution.  As illustrated in Supplemental Fig. S7, there are some light
(concentrated solute) and dark (air bubble) areas, suggesting some segregation of CsCl and air occurs
even with rapid freezing (~30 seconds).  However, this effect is less noticeable in the quickly frozen
liquid nitrogen sample (Supplemental Fig. S7), and much more pronounced in the other two freezing
methods (Figs. 3a and 3c).  Analogous findings, although using a very different experimental system,
were reported by Heger et al. (Heger et al., 2005), who found solutes were concentrated by as many as six
orders of magnitude with slow (several minutes) freezing, but only three orders of magnitude when frozen
in liquid nitrogen.
Figure 3g shows the histogram for the 1.0 mM CsCl solutions frozen using each of the three
freezing methods, as well as for Milli-Q water ice frozen in a laboratory freezer.  Unlike the images seen
in Figs. 3a through 3f, where mathematical smoothing can eliminate small structures, the histograms
include all the voxels in the sample.  As discussed in Fig 2c, water ice has two overlapping peaks,
corresponding to air bubbles (left peak) and ice (right peak).  Some voxels, shown in the "saddle"
between the two peaks, contain both air bubbles and pure water ice, and will therefore have a greyscale
value between air and ice.  The Fig. 3g histogram clearly shows how CsCl tends to be present in larger
LLR volumes in the Freezer sample, including some voxels that are almost completely composed of 2.7
M CsCl solution, with a maximum $V_{LLR}/V_{VOXEL}$ of 0.9.  This finding supports the idea of solutes
segregating to concentrated LLRs during freezing, since if solutes were precipitating and forming solid
inclusions in the bulk ice, the calculated ratio in a voxel could be higher than 1.  The fact that the ratio
gets close to, but never exceeds, 1 is consistent with our tricomponent model of air, relatively pure ice,
and concentrated LLRs with a maximum concentration of 5.4 M total solute.
The increased number of air voxels on the left end of the curve for the 1.0 mM CsCl freezer
sample represents voxels composed entirely of air. This number is larger than in the water sample,
supporting the imaging findings that the presence of solute actually increases the size of air bubbles. For
the Freeze Chamber and LN2 samples, the number of voxels containing only air is smaller, and voxels
containing air are more likely to contain at least some fraction of ice or solute. For the Freeze Chamber
sample, the histogram correlates with the images (2c and 2d), with fewer voxels containing a large
volume fraction of highly concentrated regions than in the Freezer sample. Finally, the liquid nitrogen
histogram is nearly identical to water ice, although a few voxels with concentrated solute are present (also
seen in Supplemental Fig. S7). Next, we examined the impact of solute on freezing morphology and
solute location, by replacing CsCl with Rose Bengal, a large, organic molecule (see structure in
Supplemental Fig. S8). Figure 3f (movie: Supplemental Fig. S9) shows a sample containing 1.0 mM
Rose Bengal frozen in our freeze chamber. Using 1.0 mM Rose Bengal instead of 1.0 mM CsCl (Fig. 3c)
gives a very different freezing pattern, with only a few small bubbles and no visible areas of concentrated
solute. While mathematical smoothing has likely eliminated some of the smaller structures, the overall
sample morphology is quite different than that produced by the same concentration of CsCl. Miedaner
and Miedaner and co-workers (Miedaner, 2007; Miedaner et al., 2007), using different compounds, also
found that sample morphology was highly sensitive to solute identity. Interestingly, changing solute in
our system alters not only the structure of solute inclusions, but also the size of the air bubbles. The exact
reason for the change in morphology is unclear. CsCl is more polar than Rose Bengal, and could
influence the movement of the polar water molecules into the forming ice matrix. As a relatively large
organic molecule, Rose Bengal might potentially modify the ice matrix due to its size. Finally, we note
the thermodynamically predicted final concentration of solute ions at -10° C is 5.4 M; at this
concentration CsCl should still be in solution, while a substantial portion of the Rose Bengal should have
precipitated. Whether precipitated Rose Bengal is present as solids incorporated into the ice matrix or as
precipitates in LLRs is not known.
The reproducibility of samples prepared on different days but using identical methods was quite
good, with similar patterns seen for each replicate (Supplemental Fig. S10). Each combination of
freezing method and solute gave a distinct distribution of solute and air bubbles, suggesting these two
variables have a significant impact on ice morphology in our experimental system.
Table 1 lists the calculated volume of each material domain and the total CsCl mass present,
including all sample voxels, based on segmentation described in the Methods section. As seen in the
images and histogram, the Freezer sample has the highest fraction (0.00019) of voxels containing 10% or
more LLR volume, approximately 5 times greater than the Freeze Chamber sample. In contrast, the
fraction of voxels with $V_{LLR}/V_{VOXEL}$ = 2-10% in the Freezer sample (0.003) is about half that in the Freeze
Chamber sample, and the fraction of gas bubbles appears to be less than in the Freeze Chamber sample.
However, this may be a computational artifact; voxels containing LLR next to gas bubbles will have a
greyscale value somewhere between air and LLR, and therefore may be mistakenly counted as water ice
voxels. Unfortunately, determining the magnitude of this error is difficult -requiring estimating the
surface area of both air bubbles and any adjacent LLRs to identify suspect voxels - and is beyond the
scope of this study. Because LLRs in the Freezer samples are more concentrated and appear to be more
frequently found next to air bubbles (as seen in Fig. 3b), this effect may be more pronounced in the
Freezer samples than Freeze Chamber samples. However, the number of voxels mistakenly classified as
water (or less concentrated solute) is limited to boundaries between air and LLRs and therefore small, and
should not affect the overall interpretation of results. Examining the location of the CsCl mass, more than
10% of all CsCl present in the Freezer sample is found in voxels with LLRs >10%, while in the Freeze
Chamber sample only around 1% of the mass is found in these most concentrated LLRs.  For both Freezer
and Freeze Chamber samples, about two-thirds of the CsCl mass is found in the ice compartment,
suggesting most solutes are present in very small LLR inclusions that are indistinguishable from water
ice.  For the LN2 sample, only 12% of the mass is found in detectable LLRs, with the remainder
distributed throughout the water ice.  It is also possible that the CsCl in the LN2 samples is present not as
liquid inclusions, but as solid solution within the water ice.  However, the solubility of HCl in solid ice is
(1-2) x 10-4 M (Gross et al., 1975), while the CsCl solubility in solid ice would need to be 5-10 times
greater, assuming all the CsCl is present in solid solution.  The "missing" CsCl mass here is 0.88 * 126.3
µg = 111.1 µg, or 0.66 µmol.  Assuming this solute is entirely present as LLRs with solute concentration
of 2.7 M, this equates to a total LLR volume of 0.24 µL.  The volume of pure ice (again from Table 1) is
725 µL.  Therefore, assuming the remaining CsCl is distributed equally throughout the voxels labeled as
pure ice in Table 1, the calculated average $V_{LLR}/V_{VOXEL}$ for these voxels is 0.034%, indistinguishable
from water ice in our system.  While it is possible the CsCl is present (at least partially) as solutes in the
solid ice matrix, we believe it is more likely to be present primarily as small LLR inclusions.
Additionally, we present evidence later in this paper supporting the idea that solutes are predominantly
present as LLR inclusions.
We next examined the impact of sample container on sample morphology and solute distribution
by imaging samples frozen in plastic vials instead of the glass vials we used above.  While many of the
samples discussed thus far were frozen in the laboratory freezer, most of the samples prepared in plastic
vials were frozen in the freeze chamber.  Therefore, to allow appropriate comparisons, we first present a
sample of water (no solute) frozen in the freeze chamber and compare this with previous samples frozen
in the freezer.  Milli-Q water frozen in the freeze chamber in a glass vial (Supplemental Fig. S11) gives
similar spatial distribution and somewhat smaller air bubbles sizes as a similar sample frozen in a
laboratory freezer (Fig. 2a and Supplemental Fig. S1).  However, freezing water in a plastic vial rather
than glass can make a significant difference in ice morphology, as shown in Supplemental Fig. S12.
While ice in a glass vial forms many roughly spherical bubbles, water frozen in a plastic vial using our
freeze chamber forms long vertical channels; such directional growth of air bubbles in a freezing liquid
has previously been reported (Carte, 1961).  While the reason for this morphology is not entirely clear, ,
we believe it is related to how heat is removed from the sample during freezing..  Because plastic
conducts heat more poorly than water, ice, or glass, the vial walls act as insulators, forcing heat to be
primarily removed from the bottom of the sample where the plastic vial contacts the chilled plate at the
base of the freeze chamber.  This may promote the formation of vertical air channels as the ice freezes
upwards through the sample, rather than from the walls towards the interior in the glass vial sample.
We next examine the impact of freezing in plastic for a sample containing solutes.  Supplemental
Fig. S13 shows a 1.0 mM CsCl solution frozen in the freezer in a plastic vial; compared to the similarly
treated sample frozen in a glass vial (Fig. 3a), the air bubbles and concentrated inclusions are smaller in
the plastic vial.  Interestingly, the air bubbles in the plastic vial CsCl Freezer sample do not show any of
the elongation found when Milli-Q water is frozen in a plastic vial in the freeze chamber (Supplemental
Fig. S12), which may be related to the directional heat removal in the freeze chamber.  Finally, once again
using the freeze chamber, Supplemental Fig. S14 shows 1.0 mM Rose Bengal frozen in plastic in the
freeze chamber.  Here, we see substantial volumes of LLRs and more bubbles than seen in the sample
frozen in a glass vial, but without any elongation to bubbles or LLRs.
We also performed several other experiments to examine the nature of  LLRs.  Figure 4 shows a
cross-section of microCT images of the same sample (1.0 mM CsCl, frozen in freezer) at voxel
resolutions of 16 μm (left) and 2 μm (right); the corresponding movies are in Supplemental Fig. S15.  The
areas of light grey in the lower resolution image (16 μm voxel resolution), such as the area highlighted by
the arrow, are likely areas where CsCl is present in small areas of concentrated LLRs bordered by pure
water ice, although the voxel resolution does not show these features separately.   As would be expected if
freezing water effectively excludes solutes from the forming bulk ice matrix, the right hand image shows
areas of concentrated LLRs adjacent to areas of pure water ice, supporting the idea discussed earlier that
during freezing solutes are preferentially excluded from the forming ice matrix into small areas of
concentrated solution.  The higher resolution image in Fig. 4 also shows very clearly how the solutes in
LLRs often wrap around the bubbles in the Freezer CsCl samples.
Finally, Fig. 5 (and the accompanying movie in Supplemental Fig. S16) further supports the idea that
CsCl is contained in liquid-like regions in our ice samples.  We placed a 1.0 mM CsCl sample (glass vial;
Freezer) in the microCT sample holder set at -10 °C and took images of the sample (2 μm voxel
resolution, X-Z plane) at 0, 11, and 22 hours.  The temperature gradient in the sample holder was
measured later by placing a thermocouple sensor between the glass vial and the holder wall at various
positions.  The temperature difference between the bottom and middle of the holder (approximately 1.7
cm, extending above and below the 1 cm height of the frozen sample in the vial) was 2.2 °C, resulting in a
temperature gradient of 0.13 °C mm$^{-1}$.  As seen in the three images, over the 22 hours of this experiment
the bright areas of CsCl move in the direction of the temperature gradient, towards the warmer top of the
vial, at a rate of approximately 10 μm h$^{-1}$. (i.e., 7.7 μm h$^{-1}$/(K$^{-1}$ cm$^{-1}$)).  In many cases, the solutes appear
to be migrating around the surfaces of air bubbles, which are visible as darker grey spheres.  While the air
bubbles appear to remain stationary in the ice matrix, with an estimated maximum migration rate of 0.15
μm h$^{-1}$/(K$^{-1}$ cm$^{-1}$), the CsCl moves.  Solutes are excluded from the forming ice matrix during freezing
(Hobbs, 1974; Petrenko and Whitworth, 1999); here, it appears the solutes are present as a concentrated
liquid-like solution, which can migrate either along the boundaries between air bubbles and the bulk ice,
or possibly by melting into the bulk ice itself (Notz and Worster, 2009).  While we cannot rule out the
possibility that the migrating solutes might be present as solid salt crystals, as seen in other work for ice
under a temperature gradient (Light et al., 2009), the moving solutes in our images appear to be in liquid-
like regions.  Previous studies have found bubbles migrate in a temperature gradient at rates of around
1.5-3 μm h$^{-1}$/(K$^{-1}$ cm$^{-1}$) (Dadic et al., 2010), while brine inclusions move at around 10 μm h$^{-1}$/(K$^{-1}$ cm$^{-1}$)
(Light et al., 2009).  While our results support the idea of brine moving faster than bubbles, the relative
rates in our experiments seem much different (with the bubbles moving slower and the brine moving
faster) than suggested by previous literature.  However, the earlier studies were done in systems
containing either bubbles or brine inclusions, not both; as noted by Light ((Light et al., 2009), "The effect
of included gas bubbles on brine migration has not been studied."
## 4.  Implications and Conclusions
Using microCT we directly visualized the locations of solute, gas, and bulk ice in laboratory-
prepared ice samples.  While the chemical concentrations we used are higher than those in clean polar
samples, because of the substantial morphological differences seen between pure ice samples and solute-
containing samples, we expect that solutes in natural snow and ice might sometimes have important
impacts on sample morphology, including the location and sizes of liquid-like regions and air bubbles.
Highlighting the sensitivity of ice structure to freezing conditions, we found a large difference
between samples prepared at freezing temperatures in an upright freezer (where the sample was
surrounded by cold air) versus our custom-built freeze chamber (where the sample sat on a cold plate).
Samples frozen in liquid nitrogen, as expected, did not have the large air bubbles and LLR inclusions
found in Freezer or Freeze Chamber samples; nonetheless, we did find some evidence for the segregation
of solutes into LLRs, even with the fast freezing of liquid nitrogen.
In addition to freezing conditions, the choice of solute (either Cesium chloride or Rose Bengal)
also impacted the ice sample structure differently; CsCl yielded larger air bubbles and solute inclusions
compared to Rose Bengal.  While the observed variations in the locations and sizes of solute inclusions
might be expected for solutes of different polarity and size, the influence of solute on bubble morphology
is more surprising.  CsCl samples frozen in our laboratory freezer showed large LLRs, often wrapping
around air bubbles.  While QLLs at the surface ice-air interface of ice or snow are obviously in contact
with atmospheric oxidants, the preferential collocation of internal LLRs and air bubbles represents a
previously unrecognized air-ice interface.  Depending on the chemistry occurring at this interface, the
bubbles might be a source of oxidants and other gas-phase chemicals to internal solutes, and might have
significant impacts for chemical transformations under certain conditions.
Our results here suggest that subtle changes in the preparation of laboratory ice samples can have
significant impacts on the location of solutes in samples, requiring careful and consistent sample
preparation to ensure meaningful results.  Ideally, researchers would directly evaluate the location of
solutes for each sample preparation method, as we have done here; we recognize, however, this is a
significant undertaking and not possible for every laboratory to do.  Beyond the impacts on laboratory
science, our work here may be able to help  guide further investigations to understand the driving forces
shaping snow and ice structures in the natural world, as well as investigations of the rate of chemical
reactions in various compartments in snow and ice.

**Supplemental Information**
Supplemental information is available at http://dx.doi.org/10.1594/PANGAEA.855461.

**Acknowledgements**
We gratefully acknowledge thorough and insightful comments from Hans-Werner Jacobi, Sönke Maus,
and one anonymous reviewer.  We thank Doug Rowland for microCT imaging assistance, David Paige
(Paige Instruments) for constructing the temperature-controlled microCT sample chamber, and Bill
Simpson and Peter Peterson for useful conversations and suggestions. We are grateful for funding from
the National Science Foundation (grant CHE-1214121).

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

| Sample | Initial solution volume (µL) | Total CsCl mass (µg) | Volume (mm³) [a] | | | | Volume Fraction [a, b] | | | | CsCl Mass Fraction [a, c] | | |
|---|---|---|---|---|---|---|---|---|---|---|---|---|---|
| | | | Gas | Water ice | LLR 2-10% | LLR >10% | Gas | Water ice | LLR 2-10% | LLR >10% | Water ice | LLR 2-10% | LLR >10% |
| MillQ water | | | | | | | | | | | | | |
| Freezer | 500 | 0 | 5.96 | 430 | 0 | 0 | 0.014 | 0.986 | 0 | 0 | -- | -- | -- |
| Freezer, degassed | 500 | 0 | 3.23 | 432 | 0 | 0 | 0.007 | 0.993 | 0 | 0 | -- | -- | -- |
| 1 mM CsCl | | | | | | | | | | | | | |
| Freezer | 750 | 126.3 | 5.07 | 716 | 2.35 | 0.141 | 0.007 | 0.990 | 0.003 | 0.00019 | 0.651 | 0.233 | 0.116 |
| Freeze chamber | 500 | 84.2 | 5.55 | 473 | 2.67 | 0.0176 | 0.012 | 0.983 | 0.006 | 0.000037 | 0.640 | 0.346 | 0.014 |
| Liquid nitrogen | 750 | 126.3 | 0 | 725 | 1.50 | 0 | 0 | 0.998 | 0.002 | 0 | 0.879 | 0.121 | 0.000 |

Table 1. Sample volumes and fractions by material type.

[a] "Gas" is defined as having a greyscale value of < 3996, "Water ice" is defined as containing < 2% liquid-like region (LLR), "LLR 2-10%" is water ice containing an LLR fraction of between 2 and 10%, and "LLR > 10%" is water ice containing > 10% LLR. The original sample volume (either 500 or 750 µL) is not fully captured in the volumes reported here. The segmentation process eliminates some of the lower part of the sample, reducing the reported volume somewhat.

[b] Fraction of imaged sample volume (not initial solution volume). See text for details.

[c] Fraction of total CsCl mass present in each domain. Because the mass of CsCl present in the water ice compartment could not be determined directly, we assumed any mass not present in either the LLR 2-10% or LLR >10% domain is present in the water ice domain.

**Figure Captions**

Figure 1.  Radiodensity of pure water (red open squares, three data points) and of aqueous solutions
containing CsCl (blue triangles).

Figure 2.  Reconstructed images (*a* and *b*) and histogram (*c*) of water ice samples frozen in a laboratory
freezer, imaged using microCT (~16 μm voxel size) and segmented to show air bubbles (light grey) and
the bulk ice matrix (darker grey).  The glass sample vial is not shown.  The ice in panel *a* was made using
air-saturated water, while that in *b* was made with water degassed with helium for 30 min before freezing.
Panel *c* shows the distributions of the radiodensities within the two samples, expressed as the fraction of
each voxel that would be occupied by a liquid-like region (LLR) assuming the total solute concentration
is determined by freezing point depression (i.e., 5.4 M at – 10°C; (Cho et al., 2002)).

Figure 3.  Reconstructed images and histograms of ice samples frozen using three freezing methods and
with two different solutes.  Samples were imaged using a ~16 μm voxel size and segmented to show air
bubbles (light grey), the bulk ice matrix (darker grey), voxels where $V_{LLR}/V_{VOXEL}$ is between 2 and 10%
(orange) and where $V_{LLR}/V_{VOXEL} > 10\%$ (red).  The sample vial is not shown.  a) 1.0 mM CsCl solution
frozen in freezer.  b) magnification of the area in a) identified by the dashed red square.  c) 1.0 mM CsCl
solution frozen in freeze chamber (without metal plates).  d) magnification of the dashed-line area of c).
e) 1.0 mM CsCl solution frozen in liquid nitrogen.  No air bubbles or inclusions are visible at this scale.
f) 1.0 mM Rose Bengal solution frozen in freeze chamber.  g) histogram showing distribution of voxel
counts for the CsCl and Milli-Q water ice samples shown above: water ice frozen in freezer, black dotted
line; 1.0 mM CsCl frozen in LN2, orange line; 1.0 mM CsCl frozen in freezer, blue line; 1.0 mM CsCl,
frozen in freeze chamber, green line.  The inset shows an expanded view from $V_{LLR}/V_{VOXEL} = -0.1$ to 0.1.

Figure 4.  Side-by-side micro CT cross sections of the same sample (1.0 mM CsCl, frozen in laboratory
freezer) imaged at approximately 16 μm (panel a) and 2 μm (panel b) voxel sizes.  Lighter tones indicate
areas of higher radiodensity, i.e., higher solute amounts.  The scale bar applies to both images.

Figure 5.  Vertically sliced X-ray images of a 1.0 mM CsCl ice (laboratory freezer, voxel resolution ~ 2
μm) after 0, 11, and 22 hours in the CT sample chamber.  Lighter tones indicate areas of higher
radiodensity (e.g., greater CsCl amounts).  Air bubbles are visible as darker gray spheres.  The
temperature of the sample holder was set at -10 °C, but the top of the sample was approximately 1.3 °C
warmer than the bottom, corresponding to a temperature gradient of approximately 0.13 °C mm$^{-1}$.
Arrows highlight two of the areas where CsCl moves along the direction of the temperature gradient,
from colder to warmer.

Figure 1

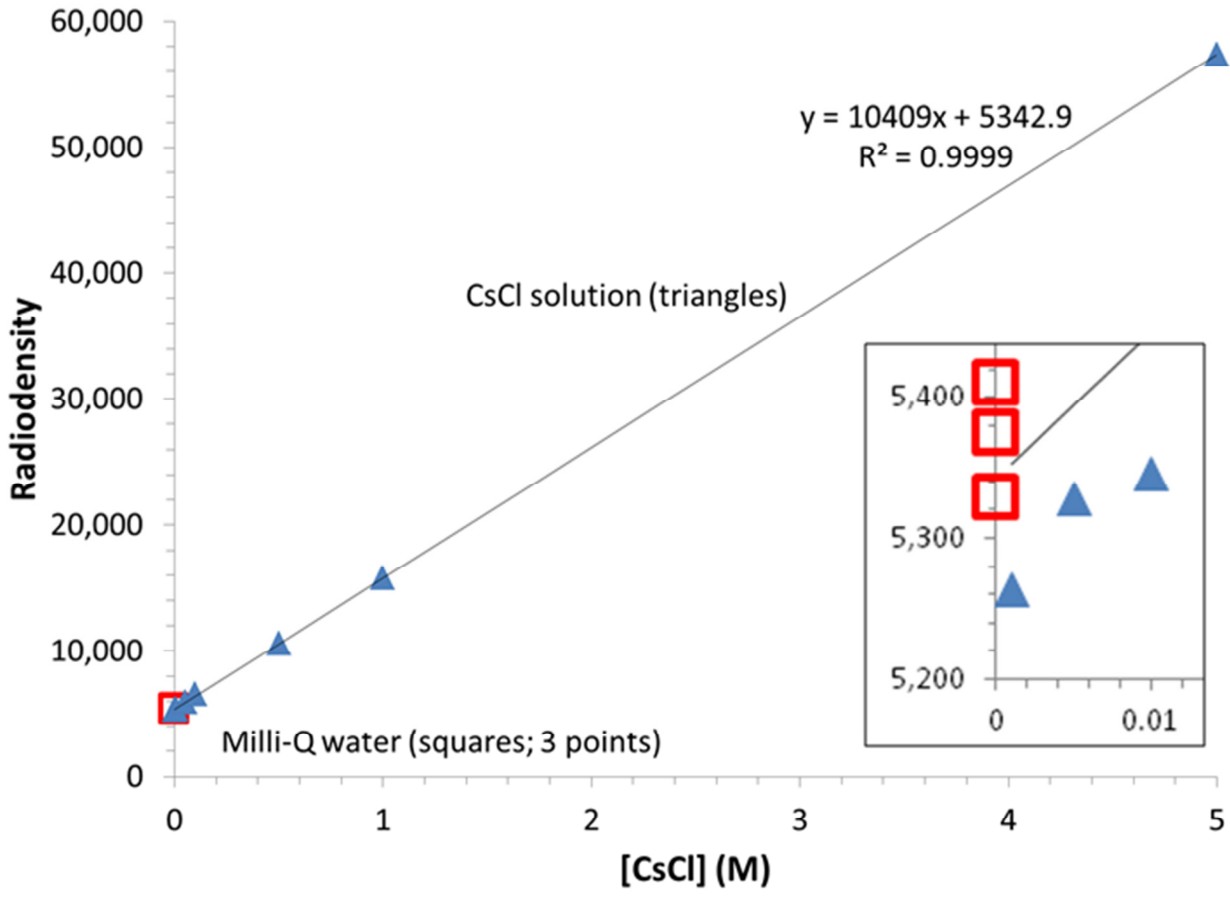


Figure 2

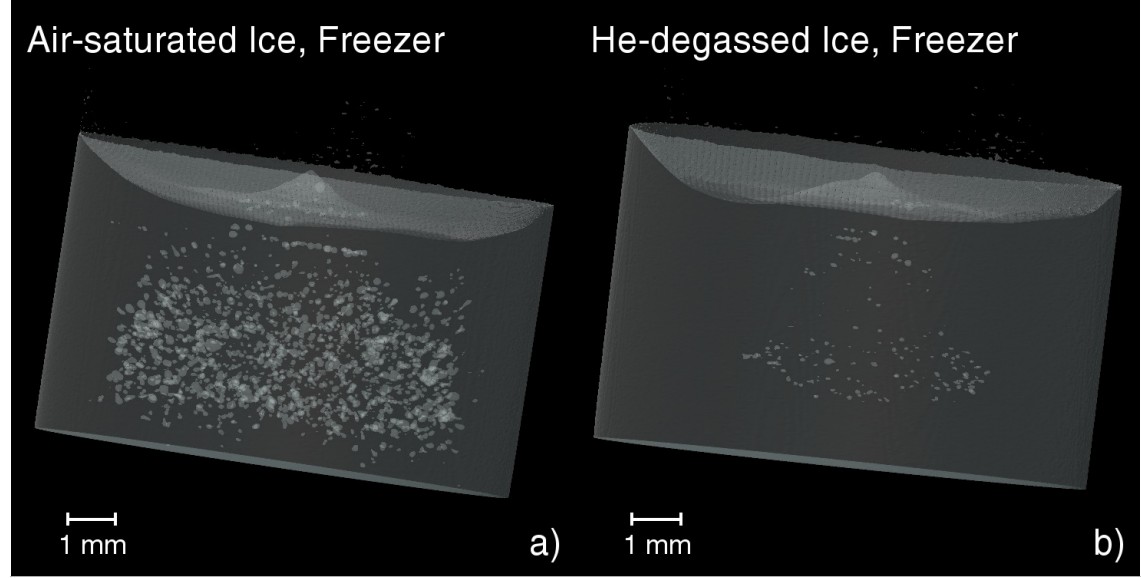


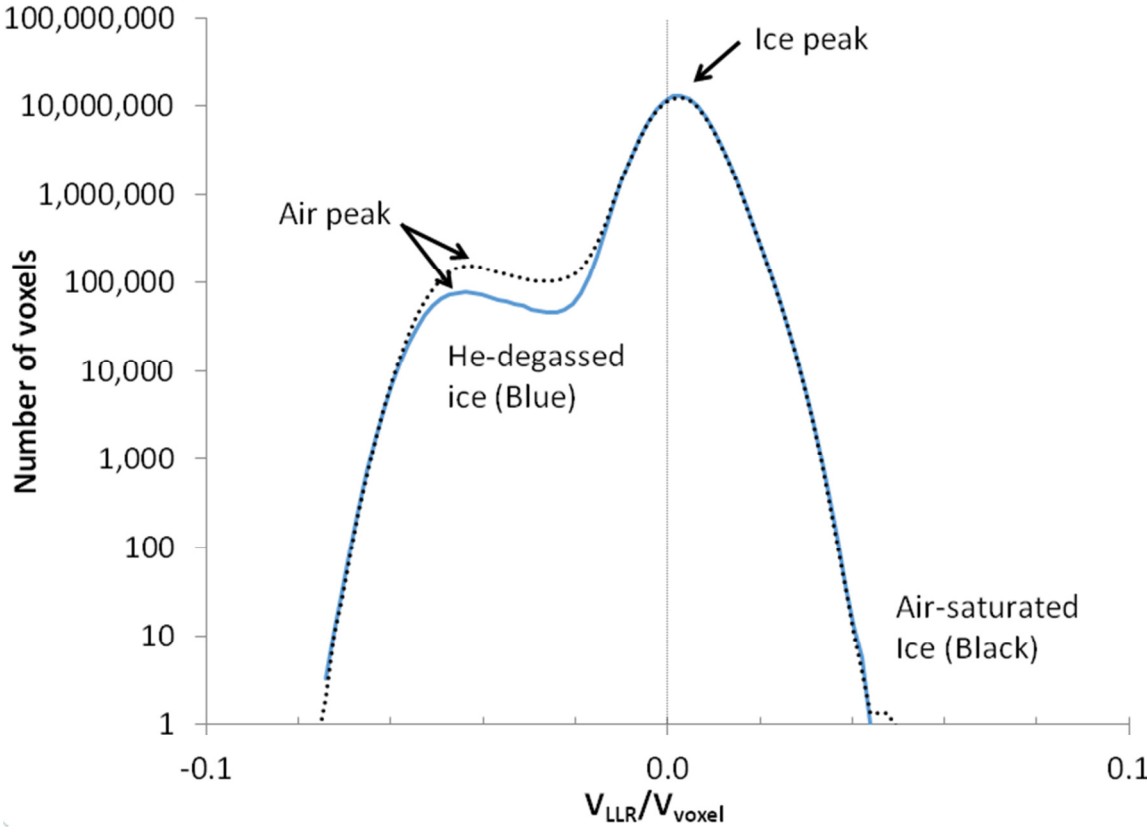


Figure 3

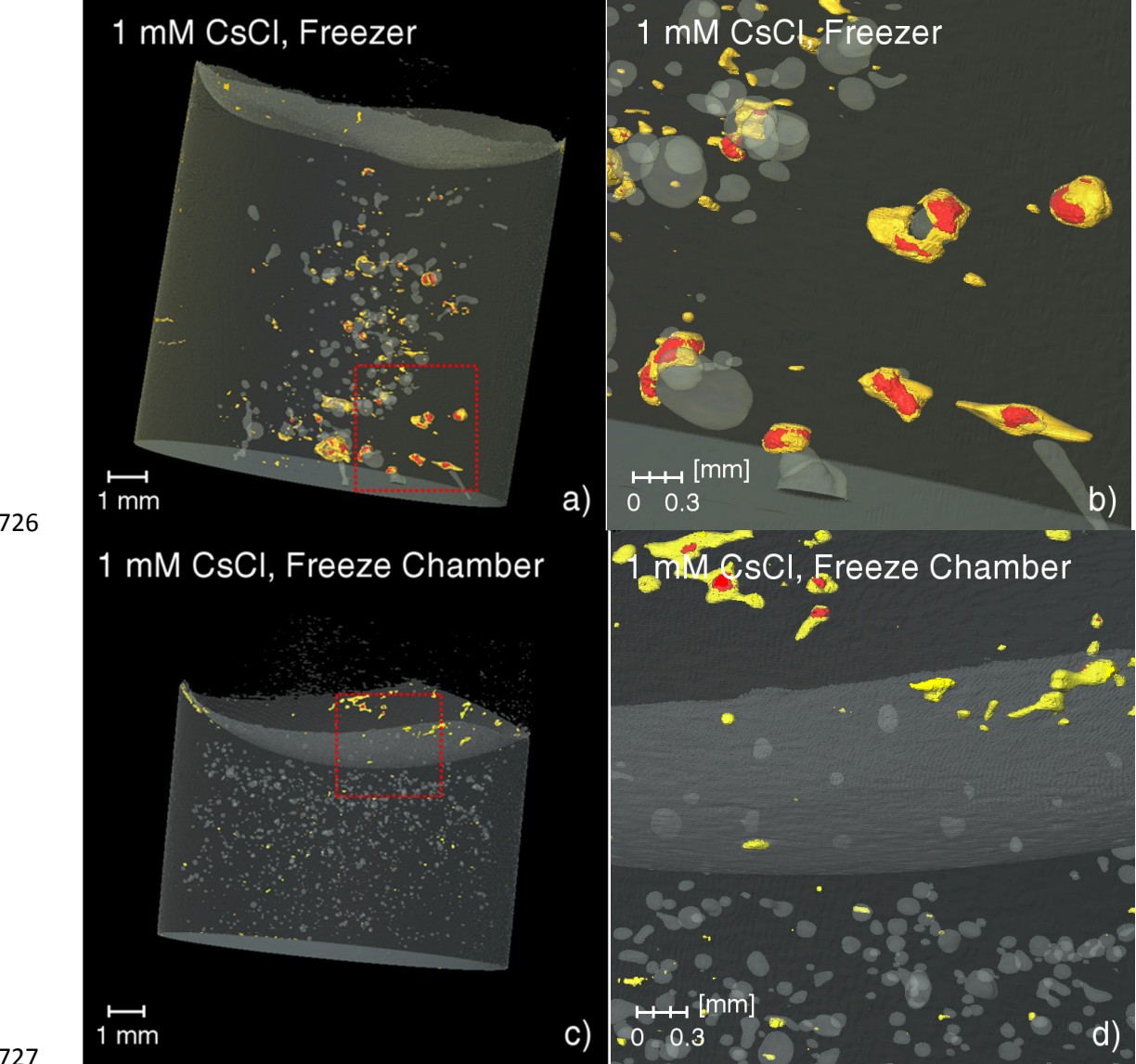

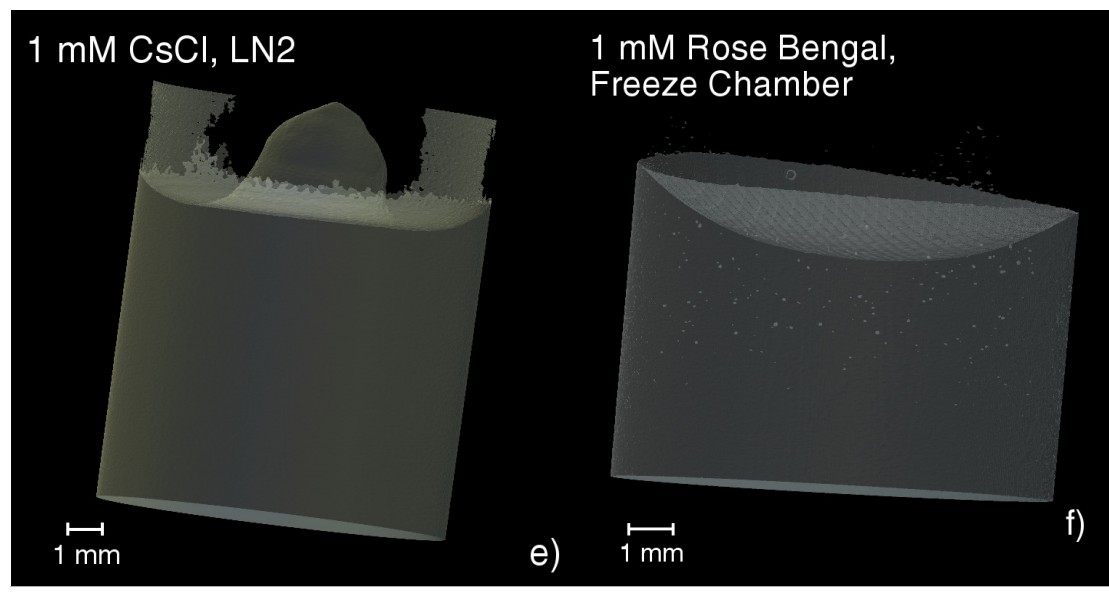

1 mM CsCl, LN2

1 mM Rose Bengal, Freeze Chamber

1 mm      e)

1 mm      f)


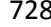

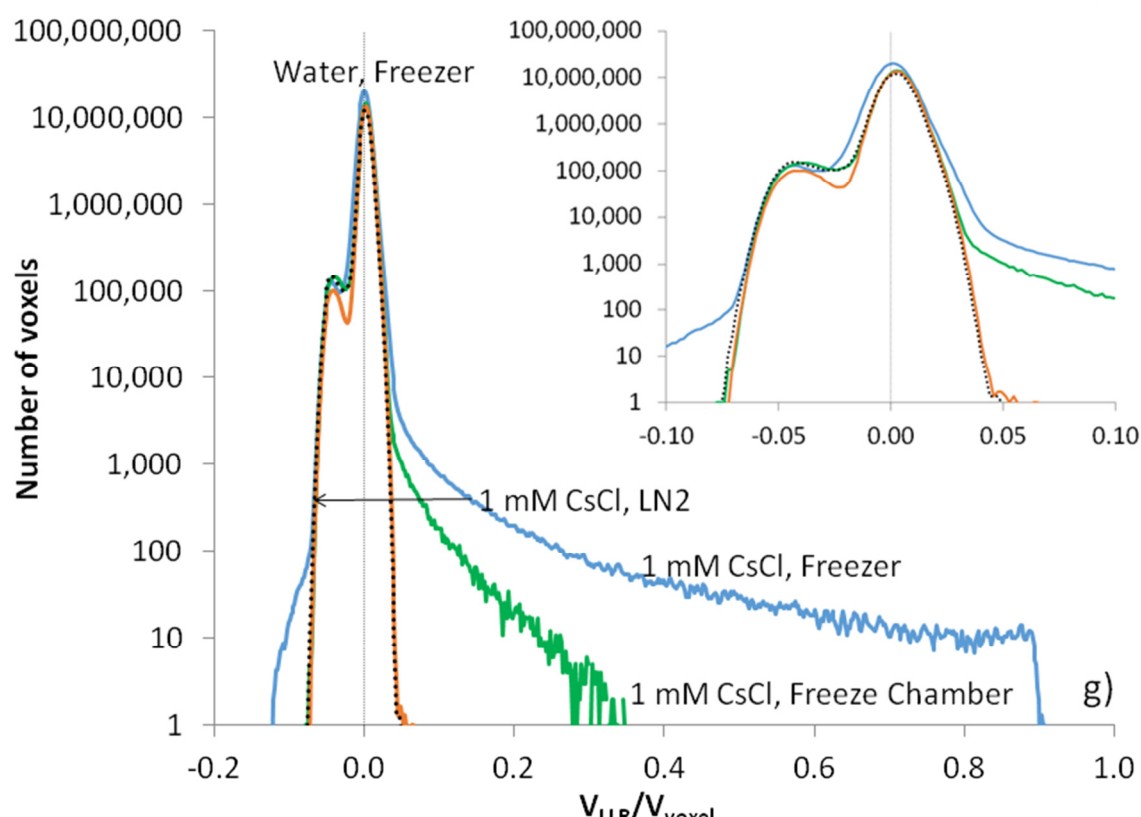


Figure 4

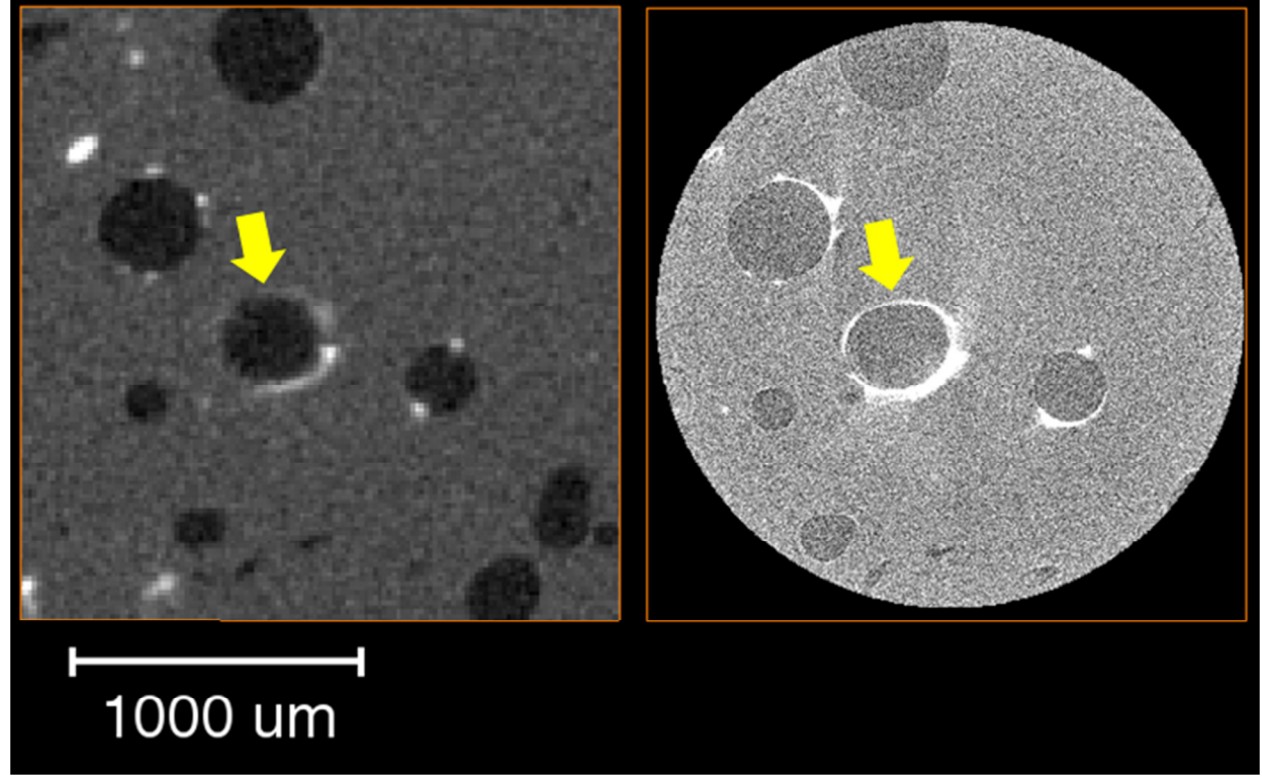


Figure 5

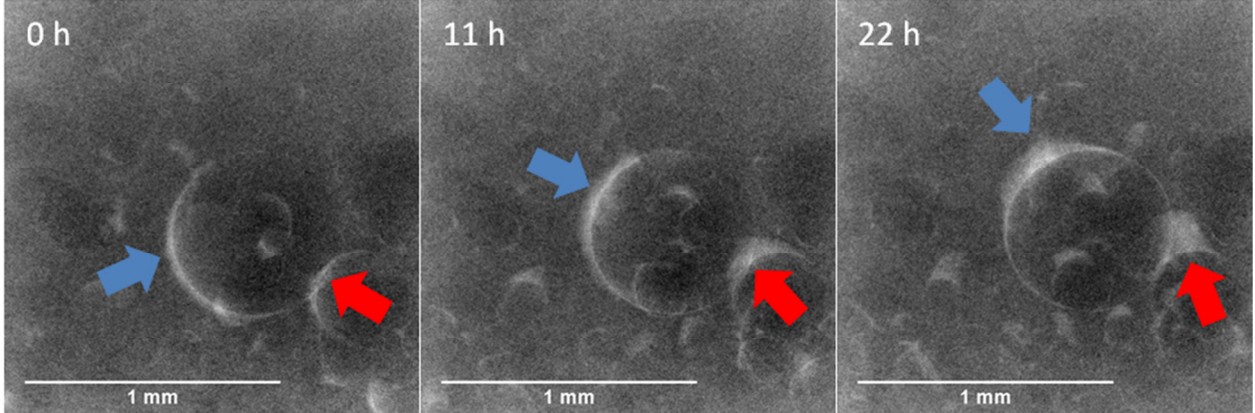
