# Peer review of "Direct visualization of solute locations in laboratory ice samples"

_The Cryosphere, 2015_

## Referee Comment (RC1) · H.-W. Jacobi (Referee) · 10 Feb 2016

The authors present an experimental study investigating the location of impurities in ice samples produced in the laboratory from aqueous solutions applying different freezing procedures. The ice samples containing either CsCl or 4,5,6,7-tetrachloro-2',4',5',7'-tetraiodofluorescein (or Rose Bengal) were analyzed using $\mu$-computed tomography, where the presence of air bubbles and the concentrations of the impurities were related to the absorption of the applied x-ray radiation. The ice samples were generated by placing liquid solutions in a freezer leading to a freezing from the exterior to the interior or by freezing from the bottom to the top or by immersion freezing with liquid nitrogen. The presented results indicate important differences concerning the distribution of the impurities in different compartments of the ice samples. The authors distinguish for the condensed phase two compartments: the solid ice matrix with low

impurity concentrations and liquid like regions (LLR) with aqueous solutions containing impurities at the solubility limit of 2.7 M in the case of CsCl. The $\mu$-CT images allow further identifying if the LLRs are located inside the solid matrix or if they are located at an interface with air bubbles. The distribution of CsCl in the ice samples are not surprising: immersion freezing leads to larger fractions of the impurities incorporated in the solid ice matrix, while the slower freezing enables a more significant formation of LLRs. These experiments constitute the first experimental evidence of the impact of the freezing method on the impurity distribution in ice samples created in the laboratory. This technique has the potential to constitute a new standard for the characterization of ice samples containing impurities that are used for laboratory experiments concerning chemical reactions in ice and snow. Such reactions are important in polar regions, where they contribute to the formation of reactive nitrogen oxides and halogens inside the snow and the subsequent release to the atmosphere. However, the exact mechanisms of the reactions and how they are modified in the different compartments of the snow or ice grains are currently not well understood and can only be studied in laboratory experiments if the impurity distribution in the samples is known. Therefore, the manuscript reports important new findings and techniques that deserve publication in The Cryosphere. Nevertheless, before the publication of the manuscript I suggest addressing the comments described below.

Comments:

Page 8: The authors propose that observed radio densities are composites of the densities of pure water ice and region with a solute ion concentration of 5.4 M, which is related to freezing point depression. This number directly affects all calculations using equation 1. How do uncertainties in the concentration translate into errors of the results? What about the error of the slope of the calibration curve shown in Fig. 1 and the errors of the radio densities for air and ice? I believe a more detailed discussion of the experimental and statistical errors and how they impact the results is needed.

The authors claim that the maximum concentration of 5.4 M in the LLR is well below

the CsCl solubility. However, its solubility at -10 °C is not known. Are there any measurements of the temperature dependence of the CsCl solubility indicating that even at -10 °C it may not be below the assumed maximum concentration? If not, could the authors determine the solubility with the $\mu$-CT using saturated solutions at different temperatures?

Page 8: The authors distinguish regions with LLR volumes either smaller or larger than 10 % of the total volume. In my opinion, this is only a gradual difference. What is the reason to identify these two categories?

Page 9f: According to the authors the degassing of the samples with helium leads to a reduction of the volume of the gas bubbles by 50 % compared to the air-saturated samples. I assume that degassing the aqueous samples with helium would only lead to replacing the dissolved nitrogen and oxygen by helium without modifying the gas volume. The degassing is a standard procedure for laboratory experiments. Is there any evidence in the literature that the volume of dissolved gas is actually reduced by a factor of 2 by degassing with helium? This may depend on the solubility of the gases nitrogen, oxygen, and helium. Did they author verify if the solubility of these different gases can explain such a difference in the dissolved gas volume?

Page 13: The authors claim that the obtained results were robust and reproducible. However, Table 1 appears to show only results from one sample for each freezing method. I recommend including data of all experiments (for example in an additional table in the Supplement) and in Table 1 average numbers including some statistical information to provide information on the uncertainty of the experimental results.

Table 1: It remains unclear to me how the numbers for example of the CsCl Mass Fraction are calculated. According to my understanding the observed radio densities for each voxel can be translated into a CsCl concentration for each voxel using the calibration curve in Fig. 1. With the known voxel volume the total mass of CsCl for each volume can be calculated. The sum of the CsCl mass for the three material types

[Figure]

(water ice, LLR 2-10%, LLR > 10 %) would give the total mass of CsCl present in each material type. In this way an absolute CsCl mass would be obtained that can then be used to calculate fractions if compared to the total CsCl mass in all three material types. Is that the procedure that was applied? If yes I recommend presenting absolute CsCl mass for each material type instead of mass fractions.

Page 8, Fig. 2 and 3: The authors state that the radio density of air is 3996. Using this value and RDice = 4848 leads to a VLLR/VVoxel ratio of -0.034. How is it possible that a significant number of voxels can have VLLR/VVoxel ratios below this value? In fact, shouldn't be the segmentation: Radio density up to 3996: voxel with only air; radio density between 3996 and 4848: ice with air bubbles; radio density above 4848: ice with CsCl? What happens with voxels that include at the same time air bubbles and LLR and both smaller than the resolution of the voxel? Do they give an average radio density signal that let them appear as solid ice without impurities?

Page 14f: The authors describe some results of their experiments using plastic vials presenting only some videos and pictures in the supplementary material. If the authors do not present a quantitative analysis like for the experiments with the glass vials, this remains more or less anecdotal and can be deleted.

The authors present the concentrations using molarities (mol/L). Wouldn't it be better to use molalities (mol/kg) to avoid the impact of density changes on the concentrations?

I think it should be mentioned in the abstract that $\mu$-CT was used.

There is a series of publication by Heger et al. (e.g. J. Photoch. Photobio. A, 187, 275–284, 2007 or J. Phys. Chem. A, 109, 6702–6709, 2005) addressing also the location of impurities in ice samples, but using completely different techniques. I recommend discussing briefly these studies in the introduction.

Page 3, lines 13ff: "As the snowpack consolidates, chemical compounds can remain at the surface of the crystals, or become trapped internally at grain boundaries or triple

junctions": I think this sentence is somewhat misleading because the mobility of the impurity is rather linked to processes at the snow grain scale and not at the snowpack scale. Thus, it rather depends on the metamorphism than on the compaction of the snowpack.

Page 3, lines 24ff: "photon fluxes can vary . . . possibly within crystals themselves." Any evidence for this statement?

Page 4, line 6f: ". . . with solutes in QLLs somewhat surprisingly having less mobility compared to solutes in LLRs." Is that really surprising assuming that the QLL may be only a few layers of water molecules?

I recommend include at least once the correct technical terminology for the compound "Rose Bengal" in the manuscript.
* * *

---

## Referee Comment (RC2) · Anonymous Referee #2 · 21 Mar 2016

I found this paper interesting and easy to read, and agree with the authors that it is important that the effect of sample preparation method is considered when interpreting the results of analyses on laboratory samples. This article is relevant to The Cryosphere and its readership, demonstrating the results of a laboratory study on experimentally grown ice.

I would recommend that this article be published, after minor revision. I found several areas where I would have appreciated more information and/or discussion on both the choice of sample preparation methods, and the discussion of results. I have split my recommendations into General comments/Specific comments and Typographical comments.

General comments

[Figure]

Introduction/Methods

From the introduction as it is written, it is unclear to me why these three sample preparation methods have been chosen. The authors suggest that their results are relevant to solute positions in snowpack, but the sample preparation procedures here - freezing directly from a liquid solution to a (presumably crystalline) solid – do not seem to be immediately comparable with snowpack formation. Additional discussion of snowpack formation mechanisms, and/or justification of these sample preparation methods, would improve the paper, clarifying the reasoning behind these experiments. (Have these sample preparation methods been used in previous experimental studies? Does the resultant structure represent snowpack well?)

Why were CsCl and Rose Bengal chosen? Are these solutes present in real snowpack? If not, what is the reasoning behind using them, are they expected to behave like the real solutes? (And what is the evidence (with references) for this?)

The samples frozen are of different sizes (Table 1). Is it possible to compare samples of different volume? Will the greater volume of the Freezer and LN2 samples not also affect the freezing behaviour?

If I understand correctly, you make the assumption with your calibration curve that the effect of CsCl on the radiodensity of water and ice is the same. And so all results based on this method are dependent on this assumption. What is the support behind this assumption? (Page 7).

Discussion

Were there any measurements of the grain size/shape of the ice in these samples? And the position of the pores/solutes with respect to grain boundaries? Might the grain size (likely controlled by the freezing temperature and subsequent annealing) play a role in the position/size of solute pockets?

The experiments are carried out with a high concentration of solutes (with respect to

inland snows). Additional discussion as to the impact of this higher concentration on the results would be enlightening – will a lower concentration have similar results?

I find the result that solutes are often associated with air bubbles in the Freezer samples, but less so for Freeze Chamber samples, very interesting – do you have any ideas of potential mechanisms which would be working differently in the two situations?

Specific comments

Page 3 Line 27. Define PAH at the first usage.

Page 4 Line 4. "the cage effect at a given temperature". It is unclear what the cage effect is, an extra sentence explaining this would be helpful.

Line 9. Not clear what is meant by "compartments"

Page 5 Line 4. You have given example of qualitative results, (visual), is there an equivalent example for quantitative?

Line 17. What is the (potential) effect of the elevated concentration of solutes?

Page 6 Line 4 Did you consider the effect of the annealing time? "At least 1 hour" – were some samples annealed for much longer? What effect might this have/did this have on the solute/pore distribution?

Line 5 - You specify the thickness of polypropylene vial walls, you should also specify the glass wall thickness. Is this the same thickness? If it isn't, any thermal consideration as to the differences between the samples frozen in glass and plastic is not only due to the difference in thermal conductivity between glass and plastic, but also the thickness. Any difference between glass and plastic vials (even though there likely will be) cannot be evidenced by these experiments if the thickness is not the same.

Line 28 "small amounts of sample" – can you quantify this?

Line 29 "very little sample in contact with air.." – similarly, is this quantifiable?

Page 7 Line 25 – the process of solute rejection on freezing has been seen in several systems, including papers earlier than the referenced Cho paper. I would suggest further literature should be included here (and possibly in other places) to avoid over-reliance on this one article.

E.g. starting points for the sea ice system: Lake and Lewis 1970 JGeophysRes 75 3 Wettlaufer et al J. Fluid Mech. (1997), vol. 344, pp. 291-316. Other systems (metals): Worster J:Fluid Mech. (1992), vol. 237, p p . 649-669

Page 8 Line 2 – what is the temperature effect on the solubility of CsCl? I would anticipate the solubility to decrease with a decrease in temperature – so the value you quote here at 20C would not be appropriate at -10C. A difference in solubility will change the calculated masses.

A quick literature search provided me with this, but there are probably other options in the literature: Jiang et al, 2003 Indian Journal of Chemical Technology Vol. 10, 391-395

Line 5 – are there no other effects on the radiodensity than the concentration of solute? What about temperature?

Line 14 – reminder of the value used for pure ice radiodensity would be helpful here.

Line 18-21 I find this sentence complicated to understand – is there a better way of presenting the four domains – in particular the distinction between 2-10% and 10% LLRs is not clear to me without rereading several times.

Page 9 Line 2 – can you put a number on the amount of solute "lost" in this process? Quantitative idea of the effect that the threshold has?

Line 4 – why only carry out this calculation for some samples? Why not all?

Line 14-16 – The different thermal conductivities between water and glass has no impact here – even if you had two materials with the same thermal conductivities, the sample would still freeze from the outside in. Suggest supressing the reference to

thermal conductivity.

Suppl Fig 1 - The directionality of bubbles in your Supplementary Figure 1 also seems to support freezing from the exterior of the sample – e.g. Carte 1961 talks about direction of bubble formation in a temperature gradient (Proc Phys Soc 77, http://iopscience.iop.org/article/10.1088/0370-1328/77/3/327/pdf.), I'm sure there are other references as well.

Line 21 can you quantify the size of the bubbles in the two figures?

Fig 2c – adding a vertical line and label at the central point of the air peak would make the histogram easier to interpret, with an equivalent label for the pure ice peak at VLLR/VVOXEL = 0.

Page 10 Line 2 – this would read better if the sentence was inverted to mention log scale before the "clearly..." comment. E.g. "Taking into account the log scale... the volume is clearly less..."

Line 12 – What are you basing the expectations of freezing speed/direction of heat removal/position of solute inclusions on? Need a reference (or more explanation of the reasoning).

Line 21 – can you find a more technical term than "blow up"? –e.g. magnification/detail...

Line 22 – do you have a mechanism/an idea for the link between solute pocket position and bubbles?

Line 24 – The use of the word "identically" here is disingenuous, the samples are not produced identically (I agree the solutions may have been). But as the aim of the paper is to demonstrate the differences when samples are produced using non-identical methods, it would bring this message home more convincingly if you avoid referring to differently frozen samples as "identically" produced. (There is another place this happens later on as well – Page 14, Line 25).

Line 29 – "blow up" – as above.

Page 11 Line 1 – "surprisingly different" – are these morphologies repeatable? To me it is not that "surprising" that a difference in freezing front gives a difference in solute distribution – maybe a different word than "surprisingly" could be used – I agree that it is interesting that this happens and that you have been able to observe it. Also, you are comparing samples of different size, this will cool/freeze differently regardless of environment.

Line 10 – Why would a freezing front process only affect the solutes and not air bubbles? I would be interested in an expansion on a theory for the mechanism for this.

Line 23 – "No air bubbles..." I don't agree there is nothing in FigS6 – I see one inclusion/something in Fig S6.

Fig 3g – (g) label appears twice. Labelling of individual curves is unclear.

Line 28 – how does histogram show that voxels contain concentrated solutes?

Page 12 Line 5 – If the effect is barely visible, does that necessarily mean it is not there/definitely not important?

Line 12 – where are the concentrated solutes in Figure 3f? It is not clear in either Fig 3f or FigS9 where these are – there seem to be only air bubbles.

Line 17-19 – this sentence is poorly worded ("much different than ...") . Can you expand on the reasons why the ice matrix would be modified? There needs to be a reference for this.

Line 24 – precipitates in LLRs? – but these LLRs are not visible in Figure 3f or Supplementary Figure 6? So how could there be precipitates within them?

Line 25 – it would improve the flow of this section if the discussion about the histogram was combined with earlier discussion (Page 11, line 7-8) as the earlier part seems truncated and unfinished.

Page 13 Line 22/Fig S10 – do you only have replicates for the freeze chamber? It would be interesting to have the equivalent histograms for each method.

Line 24 – You cannot say that the "two variables [freezing method and solute] are the primary factors influencing ice morphology" as you only change these variables in your experiments, so of course they are the two primary influencing factors here.

Page 14 Line 1 – what are the errors on the VLLR/VVOXEL = 2-10% for the two methods? These would be useful to determine whether the factor of 2 difference is reliable – as a difference between 0.003 and 0.006 doesn't seem large (but may still be significant).

Line 24 – If I understand correctly, LLRs are present when there is solute extruded from the crystallising ice – so why do you have LLRs in this (pure ice) sample?

Line 25 – use of "identical" confusing again.

Line 29 – "while the reason for this morphology is unclear..." the morphology of bubbles, and the effect of a temperature gradient, has been studied previously – e.g. Proceedings of the Physical Society, Volume 77, Number 3 http://iopscience.iop.org/article/10.1088/0370-1328/77/3/327/pdf.

Page 15 Line 15- Where are the "elongated solute inclusions" in Figure S14. They seem no more or less elongated than the air bubbles.

Figure 5 – is this a vertical slice? Specify this in the caption. Also - "along the direction of the temperature gradient" – specify which direction – i.e. from colder to warmer.

Page 16 Line 16 – a reference for "melting into the bulk ice" is required (e.g. Movement of brine pockets by salt diffusion - Notz D and Worster MG (2009) Desalination processes of sea ice revisited. J. Geophys. Res. Ocean., 114(5), C05006 (doi:10.1029/2008JC004885))

Line 20 – what is the concentration of solutes in clean polar samples? This should be

mentioned in the introduction as well.

Line 21 – "significant impacts" - I think significant should only be used in statistical context, so here "important" or similar would be more appropriate. There should also be more discussion on how this assumption that a lower concentration will have the same effect was made.

Page 17 Line 3 – this sentence ("Cesium chloride...") starts abruptly, would be improved with an introductory couple of words.

Line 14-15 "Our work here can help guide further investigations to understand the driving forces shaping snow and ice structures in the natural world, as well as the rate of chemical reactions in snow and ice." – it is not clear to me how this work helps to understand the driving forces etc, nor how there is any (direct) impact on studies into the rate of chemical reactions.

Line 16-19 "At the same time, our results suggest subtle changes in the preparation of laboratory ice samples can have significant impacts on the location of solutes in samples, requiring careful and consistent sample preparation to ensure meaningful results." - this part is the important conclusion from this work.

I would therefore suggest removing/reducing/rephrasing the first half of this paragraph, and expanding on the second half.

Supplementary info

It would be more coherent in there was only one format for videos and one for images in the supplementary information.

It would be useful to have brief captions/explanations of supplementary images and videos in the supplementary information, to avoid having to search through the main text for the information.

The colours marked "orange" in the text do not appear orange in the videos (at least

on my screens), they are yellow.

Typographical errors

Pg 3 Line 6 "to form hydroxyl radicals" Pg 5 Line7 – missing word? "was produced from..." Pg 5 Line 10 - repetition of "its" Pg 8 Line 17 "VVOXEL" should be in capitals Pg 11 (22-23) "Results for a 1.0mM CsCl sample.. are shown in Fig. 3e"

Figure 4 – the blue arrows are not visible when printed in black and white.

---

## Referee Comment (RC3) · S. Maus (Referee) · 7 Apr 2016

**Review of tc-2014-88**

This is a review of the Cryosphere Discussions manuscript tc-2015-197 *Direct visualization of solute locations in laboratory ice samples* by Theodore Hullar and Cort Anastasio.

**General comments**

**Summary**

The paper presents and discusses results from X-ray tomographic imaging of aqueous solutions frozen in small containers in the laboratory. With two solutions, cesium chloride (CsCl) and Rose Bengal solution, three different freezing methods were used: (i) freezing by putting containers in a normal freezer, (ii) unidirectional bottom-up freezing with containers placed on a cold plate and (ii) putting small vials into liquid nitrogene. The frozen samples are imaged my X-ray microtomography, mostly at a voxel size of 16 $\mu$m, to obtain 3-d greyscale images of X-ray transmission. After segmentating the greyscale images into different classes that reflect solute and air or gas content, the authors discuss these images qualitatively in terms of distribution of solute inclusions and air bubbles. The authors also perform a quantitative analysis of the distribution and content of solute in liquid like regions (LLRs) versus solute incorporated in the solid ice matrix, as well as some observations on the movement of liquid inclusions. The authors conclude that the work shows that *the structure of laboratory ice samples, including the location of solutes, is sensitive to freezing method, sample container, and solute characteristics, requiring careful experimental design and interpretation of results*. The work is proposed to enhance our understanding of solute segregation in natural snow and ice, as well as of *the air-ice interface and liquid-like regions within the ice matrix*.

I agree with two other referees that the paper is relevant for ´The Cryosphere' and its readership, and also mostly with their comments. I would like to add comments on two major issues on which the paper in my opinion needs improvement. First, I encourage the authors to improve review and referencing of the published background on ice observations and solute redistribution during freezing, to be included in the introduction and discussion of their own observations. Second, I think that there is potential for improvement in the quantitative analysis of the 3-d images, and a critical discussion of the method and results. In my comments I will give literature examples that I hope will help the authors to improve their analysis and presentation.

**Main concerns I-III**

I. Background - ice, freezing and solute inclusions

1. In the literature on ice physics and chemistry there are several books that include fundamental discussions of ice structure and solute distribution during freezing but none of these is mentioned. I recommend to have a look into the literature (e.g., Shumskii, 1955; Hobbs, 1974; Lock, 1990; Petrenko and Whitworth, 1999; Pruppacher and Klett, 1997), and possibly cite from there.

2. It is well known that during the freezing of saline solutions most solute is rejected into the remaining mother solution and not incorporated into the solid ice matrix (e.g., Hobbs, 1974; Petrenko and Whitworth, 1999). This fact should be more clearly

mentioned in the text. The authors for example write (P16, L13-16) *While the air bubbles remain stationary in the ice matrix, the CsCl moves, consistent with the idea that solutes are present as a concentrated liquid-like solution, which can migrate either along the boundaries between air bubbles and the bulk ice, or possibly by melting into the bulk ice itself.* Such formulation indicates that solute rejection during solidification of water is only an *idea* rather than a fact under most conditions.

3. I am also missing background literature on observations of solute inclusions in ice. For example, already Quincke (1905) has described the morphology and distribution of liquid inclusions of ice grown from saline solutions, and there is much more information in the books on ice physics mentioned above. As an example, Shumskii (1955) describes the solute distribution in ice as (p.180): 'The distance between neighbouring interlayers of inclusions in a crystal decreases with increasing concentration of impurities in the remainder mother solution; often this distance is as much as 35-45 $\mu$ with inclusions 8-15 $\mu$ thick'. Such information is certainly relevant for the discussion and interpretation of the results in the present paper.

4. An idea of the expected microstructure and potential separation of solute inclusions and air bubbles may be obtained by consulting published work based on thin section analysis of frozen solution or pure water droplets (e.g., Hallett, 1964; Rohatgi and Adams, 1967). For example, a useful information from these studies is the dependence of dendrite or plate spacing on freezing velocity: The faster the freezing, the smaller the spacing of ice plates and solute inclusions, which normally implies smaller dimensions of solute inclusions. For the present study this may affect the detectability of solutes, especially for the samples frozen very rapidly in liquid nitrogene-based freezing.

5. Earlier basic work on solubility of ions in the solid matrix as well as solute partitioning at a freezing interface (e.g., Tiller, 1963; Gross et al., 1975, 1977) should be mentioned and discussed. Such information is particularly important when it comes to the quantitative analysis and discussion - see my comments below.

6. How is the freezing point depression the authors assume for CsCl (2.7 M CsCl at -10 ℃) computed, or on which reference is it based? E.g. according to Pruppacher and Klett (1997) (p. 125, Fig. 4-12) one might expect that a value of 3.1 to 3.3 M is a more realistic value at -10 ℃. While such a change in equilibrium concentration would affect the estimates of the volume of LLR from the greyscale images, it would not affect the solute content within liquid inclusions. However, it would decrease the maxium solute content in voxels in the histogram, and thus may give hints on the proper estimation and possible rescaling of equation (1).

7. X-ray tomography of solutions frozen in the laboratory has been performed earlier Miedaner (2007); Miedaner et al. (2007) and may be compared to the present results.

II. Image segmentation and quantitative analysis of solute content and locations

1. The proposed segmentation approach is based on equation (1) on page 8 assuming 2.7 M equilibrium concentration CsCl at -10 ℃. Please have a closer look in the literature to evaluate the uncertainty of this estimate.

2. The choice of a cutoff at $LLR = 2\%$ to estimate the solute content in liquid like regions seems somewhat arbitrary. As the pure ice histogram in Fig. 2c extends to slightly above $LLR = 4\%$, rather such a cutoff would be more consistent. At least would a $LLR = 4\%$ cutoff define a lower bound of the solute content in liquid inclusions.

3. As the authors correctly point out, the solute content will be underestimated due to the possibility of mixed air and solute pixels, which may then have radio densities between brine and air, and thus be classified as ice. Can this bias be estimated? An approach to place a bound on this bias could be to count the surface voxels of the air bubbles and assume that these contain CsCl brine and air. How would this affect the results in Table 1?

4. In the histogram in Fig. 3 the maximum volume fraction of liquid is roughly 0.9. However, if calibration and equation (1) would be correct I would expect that the maximum value should at least be 1 (due to expected noise even larger), provided that there are at least some liquid inclusions that exceed the volume of a voxel with side length $16\mu$m. While this may not be the case, I would expect that it would very likely be the case for high resolution imaging with $2\mu$m voxel size. How does such a histogram look like, and may it be used to improve the calibration in eq. (1)?

5. According to Table 1 the solute content classified in solute inclusions is 12-35 %, and the remainder is concluded to be incorporated in the ice matrix. How does this compare to expected solubility limits in the solid? E.g., Gross et al. (1975) suggested a solubility limit of 1-2 $\times 10^{-4}$ M for HCl in the solid ice matrix. The result from the authors calculations (65 to 88% of the initial 1 mM CsCl in the ice matrix) would roughly imply a 3-9 times larger solubility of CsCl in ice. Do any studies exist that support such a high solubilty of CsCl in solid ice? If not, then this might be another indication that eq. (1) should be changed by a prefactor that gives larger liquid fractions of at least 1 at the higher end of the histogram. Again, it appears very important to present a similar analysis of high resolution images, that could solve this problem.

6. The inset in Figure 3 compares the histogram envelope around the radiodensity of ice for the Milli-Q and solute samples. It indicates that the ice peak and envelope in the histogram is slightly shifted to the right for the frozen solutions with respect to frozen Milli-Q - which is particularly apparent for the LN2 samples. Such a shift would indeed be consistent with Cs and Cl incorporated in the solid ice matrix, where they act in the same way as strong X-ray absorbers as when in liquid solution. It would be very interesting to evaluate, if it is possible to estimate the solute content in the ice matrix from this shift.

7. A statistical analysis of size distribution of slute inclusions and air bubbles would be very helpful. Such a statistics would also justify to include the results from Rose Bengal solutions, that else is given too little weight in this study.

III. Results and Discussion

1. Every paragraph in the discussion contains a reference to supplementary material, that is the discussion is based very much on the latter (S1-S16). While it is helpful to provide such material, I regard it as inappropriate to build up the discussion of a research paper on that much supplementary information. Some of this information should become part of the paper and the discussion should be rewritten.

**Specific comments**

P 4, L 23-25 –> *But to our knowledge this method has not been used to investigate the structure and solute locations for laboratory samples prepared under controlled conditions with specific solutes* - I would not call the freezing conditions *controled*, as neither cooling rates or supercooling in the samples were controled or measured.

P 4, L 29 –> *In this work we focus on cesium chloride (CsCl) as our solute. However, because a previous study (Cheng et al., 2010) found different solutes can affect freezing morphology and therefore may influence solute location, we also imaged ice containing the organic compound Rose Bengal.* I suppose that CsCl was choosen because it warrants a high X-ray contrast between ice and solute. Why was Rose Bengal choosen? Also, as the results presented are, except for a histogram in Fig. 3 as well as supplementary material, for the CsCl solutions, I would rather suggest to remove the few Rose Bengal results and notes, and rather present a systematic and quantitative comparison elsewhere.

P 5 L 1 –> *Cheng et al., 2010* - this is a reference to a study based on a rather different method, that yields the surface distribution of solutes/ions. There exist other studies that have shown the influence of solute on freezing pattern, for example the mentioned work by Rohatgi and Adams (1967). I cannot see that the cited paper is an argument to use Rose Bengal as an alternative solution.

P 16 L 13 –> *While the air bubbles remain stationary in the ice matrix, the CsCl moves, consistent with the idea that solutes are present as a concentrated liquid-like solution, which can migrate either along the boundaries between air bubbles and the bulk ice, or possibly by melting into the bulk ice itself* - First, I find it surprising, that the air bubbles remain stationary, because it is well established that air bubbles migrate in a temperature gradient at similar rates as liquid inclusions (Dadic et al., 2010). Second, some refererence on the process of brine pocket migration should be mentioned here, please have a look at Light et al. (2009) and the literature reviewed therein. Third, Light et al. (2009) also found migration for solid crystals, so the movement of solute is no proof for its liquid character.

P 16 L 23 –> *surprisingly* - considering earlier studies on the freezing of saline solutions I would not rate this as surprising.

**References**

Dadic, R., Light, B., Warren, S. G., 2010. Migration of air bubbles in ice under a temperature gradient, with application to snowball earth. J. Geophys. Res. 115, D18125.

Gross, G. W., Wong, P. M., Humes, K., 1977. Concentration dependent solute redistribution at the ice-water phase boundary. III. Spontaneous convection, chloride solutions. J. Chem. Phys. 67 (11), 5264–5274.

Gross, G. W., Wu, C., Bryant, L., Wu, C., 1975. Concentration dependent solute redistribution at the ice-water phase boundary. II. Eperimental investigation. J. Chem. Phys. 62 (8), 3085–3092.

Hallett, J., 1964. Experimental studies of the crystallization of supercooled water. J. Atmosph. Sci. 21, 671–682.

Hobbs, P., 1974. Ice Physics. Clarendon Press, Oxford, 837 pp.

Light, B., Brandt, R., S.G.Warren, 2009. Hydrohalite in cold sea ice: Laboratory observations of single crystals, surface accumulations, and migration rates under a temperature, gradient, with application to ´snowball earth'. J. Geophys. Res. 114, C07018.

Lock, G. S. H., 1990. The Growth and Decay of Ice. Cambridge University Press, Cambridge, 434 pp.

Miedaner, M. M., 2007. Characterization of inclusions and their distribution in natural and artificial ice samples by synchrotron cryo-micro-tomography (scxrt). Ph.D. thesis, Institute of Environmental Sciences, Johannes Gutenberg University Mainz, 113 pp.

Miedaner, M. M., Huthwelker, T., Enzmann, F., Kersten, M., Stamponi, M., Amman, M., 2007. X-ray tomographic characterization of impurities in polycrystalline ice. 11th Int. Conf. on Physics and Chemistry of Ice, Bremerhaven, Germany. Royal Society of Chemistry, pp. 399–408, in press.

Petrenko, V. F., Whitworth, R. W., 1999. Physics of Ice. Oxford University Press, 373 pp.

Pruppacher, H. R., Klett, J. D., 1997. Microphysics of clouds and precipitation, 2nd Edition. Vol. 18 of Atmospheric and oceanographic sciences library. Kluwer Acadamics Publ., 954 pp.

Quincke, G., 1905. Über Eisbildung und Gletscherkorn. Annalen der Physik, Leipzig 18(11), 1–80.

Rohatgi, P. K., Adams, C. M., 1967. Ice-brine dendritic aggregate formed on freezing of aqueous solutions. J. Glaciol. 6 (47), 663–679.

Shumskii, P. A., 1955. Osnovy strukturnogo ledovedeniya (Principles of structural glaciology). Dover Publications, Inc., 1964 translated from Russian, 497 pp.

Tiller, W. A., 1963. Principles of solidification. John Wiley & Sons, Ch. 15, pp. 276–312.

---

## Author Comment (AC1) · 27 Jun 2016

The author response has been uploaded as a Supplement, and consists of:

1. Responses to individual comments from reviewers 2. Updated manuscript in Microsoft Word format, with changes tracked for editor and reviewer convenience 3. Updated manuscript in Microsoft Word format 4. Word file containing captions for supplementary files

Please also note the supplement to this comment:
http://www.the-cryosphere-discuss.net/tc-2015-197/tc-2015-197-AC1-supplement.zip
* * *

---

## Author Response (AR1)

**Response to Comments**

**Direct visualization of solutes in laboratory ice samples**
Ted Hullar, Cort Anastasio
The Cryosphere (Discussion), 15 Jan 2016

**Comments from H-W Jacobi (Referee):**

**The authors present an experimental study investigating the location of impurities in ice samples produced in the laboratory from aqueous solutions applying different freezing procedures. The ice samples containing either CsCl or 4,5,6,7-tetrachloro-2',4',5',7'- tetraiodofluorescein (or Rose Bengal) were analyzed using $\mu$-computed tomography, where the presence of air bubbles and the concentrations of the impurities were related to the absorption of the applied x-ray radiation. The ice samples were generated by placing liquid solutions in a freezer leading to a freezing from the exterior to the interior or by freezing from the bottom to the top or by immersion freezing with liquid nitrogen. The presented results indicate important differences concerning the distribution of the impurities in different compartments of the ice samples. The authors distinguish for the condensed phase two compartments: the solid ice matrix with low impurity concentrations and liquid like regions (LLR) with aqueous solutions containing impurities at the solubility limit of 2.7 M in the case of CsCl. The $\mu$-CT images allow further identifying if the LLRs are located inside the solid matrix or if they are located at an interface with air bubbles. The distribution of CsCl in the ice samples are not surprising: immersion freezing leads to larger fractions of the impurities incorporated in the solid ice matrix, while the slower freezing enables a more significant formation of LLRs. These experiments constitute the first experimental evidence of the impact of the freezing method on the impurity distribution in ice samples created in the laboratory. This technique has the potential to constitute a new standard for the characterization of ice samples containing impurities that are used for laboratory experiments concerning chemical reactions in ice and snow. Such reactions are important in polar regions, where they contribute to the formation of reactive nitrogen oxides and halogens inside the snow and the subsequent release to the atmosphere. However, the exact mechanisms of the reactions and how they are modified in the different compartments of the snow or ice grains are currently not well understood and can only be studied in laboratory experiments if the impurity distribution in the samples is known. Therefore, the manuscript reports important new findings and techniques that deserve publication in The Cryosphere. Nevertheless, before the publication of the manuscript I suggest addressing the comments described below.**

**Page 8: The authors propose that observed radio densities are composites of the densities of pure water ice and region with a solute ion concentration of 5.4 M, which is related to freezing point depression. This number directly affects all calculations using equation 1. How do uncertainties in the concentration translate into errors of the results? What about the error of the slope of the calibration curve shown in Fig. 1 and the errors of the radio densities for air and ice? I believe a more detailed discussion of the experimental and statistical errors and how they impact the results is needed.**

We have expanded the discussion of the errors associated with uncertainties in the solute ion concentration the LLRs. Errors in the LLR solute concentration would scale the findings, changing $V_{LLR}/V_{VOXEL}$ for our results. There is some uncertainty in the LLR solute ion concentration, perhaps as great as 20%. This error is considerably larger than the error in the slope of the calibration curve and the radiodensities for air and ice, and we have therefore discussed it more extensively than the other sources of error.

**The authors claim that the maximum concentration of 5.4 M in the LLR is well below the CsCl solubility. However, its solubility at -10 ◦C is not known. Are there any measurements of the temperature dependence of the CsCl solubility indicating that even at -10 ◦C it may not be below the assumed maximum concentration? If not, could the authors determine the solubility with the $\mu$-CT using saturated solutions at different temperatures?**

We were unable to locate data on the solubility of CsCl at -10 °C.  However, one of the reasons we chose CsCl as our primary test compound was its high solubility in water.  According to NIH (NIH, 2015), the saturation concentration of CsCl in water is approximately 9.6 M at 0 °C and 11.1 M at 20 °C .  Assuming the saturation concentration change is linear with temperature, the saturation concentration at -10 °C would be 8.9 M, still well above the estimated LLR solute concentration of 2.7 M.  Note that 5.4 M refers to the total solute concentration; not the concentration of CsCl.

**Page 8: The authors distinguish regions with LLR volumes either smaller or larger than 10 % of the total volume. In my opinion, this is only a gradual difference. What is the reason to identify these two categories?**

We agree with the opinion of the reviewer, as shown in Figure 3g, the LLR volume in a particular voxel can range from zero to above 90%, with no particular importance associated with fractions above or below 10%.  We chose 10% as the dividing line because it allowed us to show how areas of highly concentrated solute (>10%) were more common in the freezer samples, and also spatially more likely to surround air bubbles; the choice of actual percentage was arbitrary.

**Page 9f: According to the authors the degassing of the samples with helium leads to a reduction of the volume of the gas bubbles by 50 % compared to the air-saturated samples. I assume that degassing the aqueous samples with helium would only lead to replacing the dissolved nitrogen and oxygen by helium without modifying the gas volume. The degassing is a standard procedure for laboratory experiments. Is there any evidence in the literature that the volume of dissolved gas is actually reduced by a factor of 2 by degassing with helium? This may depend on the solubility of the gases nitrogen, oxygen, and helium. Did they author verify if the solubility of these different gases can explain such a difference in the dissolved gas volume?**

There is some evidence in the literature that degassing with helium does reduce the volume of dissolved gas by around a factor of 2 (Snyder, 1983). The degree of degassing does depend on the solubility of gases involved.  Assuming a solution in equilibrium with air at 25 C, the mole fraction solubility of air (assuming a composition of 20% oxygen and 80% nitrogen) is 1.4 x $10^{-5}$, while for helium it is 7.0 x $10^{-6}$, or almost exactly half the concentration of air in the solution.  This difference does explain the observed reduction in bubble volume in the helium-degassed solution.  We have added this calculation to the paper.

**Page 13: The authors claim that the obtained results were robust and reproducible. However, Table 1 appears to show only results from one sample for each freezing method. I recommend including data of all experiments (for example in an additional table in the Supplement) and in Table 1 average numbers including some statistical information to provide information on the uncertainty of the experimental results.**

Both generating the microCT imaging, and particularly segmenting those results into air, water ice, and LLRs, requires considerable effort.  We were not able to evaluate as many samples as we would have liked, focusing instead on studying a range of sample conditions.  We have included the histograms in

Supplemental Figure S10 as a way to provide some insight into sample reproducibility by comparing the raw (unsegmented) greyscale values from three identically-prepared samples.

**Table 1: It remains unclear to me how the numbers for example of the CsCl Mass Fraction are calculated. According to my understanding the observed radio densities for each voxel can be translated into a CsCl concentration for each voxel using the calibration curve in Fig. 1. With the known voxel volume the total mass of CsCl for each volume can be calculated. The sum of the CsCl mass for the three material types (water ice, LLR 2-10%, LLR > 10 %) would give the total mass of CsCl present in each material type. In this way an absolute CsCl mass would be obtained that can then be used to calculate fractions if compared to the total CsCl mass in all three material types. Is that the procedure that was applied? If yes I recommend presenting absolute CsCl mass for each material type instead of mass fractions.**

We used essentially the process described, with one change. We first segmented the imaged sample into four materials (including air bubbles), as described in the Methods section. For the two material types containing regions of concentrated LLRs (LLR 2-10% and LLR > 10%), we determined the total volume of the material type, as well as the average CsCl concentration in that volume. From these two pieces of information, we calculated the CsCl mass present in each material type. We could not directly measure CsCl present in the bulk ice, because the radiodensity for CsCl containing voxels overlapped with the radiodensity distribution for pure water ice. However, because we knew the volume and concentration of the initial solution, we knew the mass of CsCl present in the overall system. From that information, we calculated the CsCl mass present in the bulk ice by subtracting the overall original mass from the mass present in the other two materials.

**Page 8, Fig. 2 and 3: The authors state that the radio density of air is 3996. Using this value and RDice = 4848 leads to a VLLR/VVoxel ratio of -0.034. How is it possible that a significant number of voxels can have VLLR/VVoxel ratios below this value? In fact, shouldn't be the segmentation: Radio density up to 3996: voxel with only air; radio density between 3996 and 4848: ice with air bubbles; radio density above 4848: ice with CsCl? What happens with voxels that include at the same time air bubbles and LLR and both smaller than the resolution of the voxel? Do they give an average radio density signal that let them appear as solid ice without impurities?**

Visualization of a pure material using the micro-CT does not yield a single radiodensity value, but rather a distribution of values. While 3996 is the average radiodensity of air (page 8), some values are greater, some are less. This effect is visible in Figure 2c, where the air distribution (smaller left peak) overlaps the ice distribution (larger right peak). So, for pure air voxels, half should have $V_{LLR}/V_{Voxel}$ values below -0.034, half greater. Deconvolution of these values is challenging; in fact, as the reviewer notes, a voxel containing an air bubble and LLR might well have an average radiodensity close to that of pure water, and be incorrectly categorized as such. We have attempted to discuss some of the uncertainties around this issue on pages 8 and 9. In addition, higher resolution imaging, such as Figure 4 and Supplemental Figure 7, can help resolve these "border" questions.

**Page 14f: The authors describe some results of their experiments using plastic vials presenting only some videos and pictures in the supplementary material. If the authors do not present a quantitative analysis like for the experiments with the glass vials, this remains more or less anecdotal and can be deleted.**

While we agree the results are qualitative, we do believe the plastic vial discussion could provide some useful guidance to other researchers who may not have considered the importance of sample container.

**The authors present the concentrations using molarities (mol/L). Wouldn't it be better to use molalities (mol/kg) to avoid the impact of density changes on the concentrations?**

Because the micro-CT imaging is done on a volumetric basis, we believe expressing concentrations in molarities was the most appropriate unit. We did our standard curve calibration knowing how much solute was present in a particular volume of solution. By then imaging an identical volume of pure water (or ice), we could determine how the radiodensity of that volume changed when various amounts of solute were present.

**I think it should be mentioned in the abstract that $\mu$-CT was used.**

We agree, and have revised the abstract.

**There is a series of publication by Heger et al. (e.g. J. Photoch. Photobio. A, 187, 275–284, 2007 or J. Phys. Chem. A, 109, 6702–6709, 2005) addressing also the location of impurities in ice samples, but using completely different techniques. I recommend discussing briefly these studies in the introduction.**

We have discussed these studies in the introduction as suggested.

**Page 3, lines 13ff: "As the snowpack consolidates, chemical compounds can remain at the surface of the crystals, or become trapped internally at grain boundaries or triple junctions": I think this sentence is somewhat misleading because the mobility of the impurity is rather linked to processes at the snow grain scale and not at the snowpack scale. Thus, it rather depends on the metamorphism than on the compaction of the snowpack.**

We have clarified the wording of the sentence to state "As the snowpack consolidates and snow grains metamorphose, chemical compounds can remain at the surface of the crystals, or become trapped internally at grain boundaries or triple junctions (Domine et al., 2008; Grannas et al., 2007)."

**Page 3, lines 24ff: "photon fluxes can vary . . . possibly within crystals themselves." Any evidence for this statement?**

McFall and Anastasio (McFall and Anastasio, 2016) found slight enhancement of photon flux in ice crystals versus aqueous solutions, which suggests variation within the crystals may be possible. We have revised the text to include this reference.

**Page 4, line 6f: ". . . with solutes in QLLs somewhat surprisingly having less mobility compared to solutes in LLRs." Is that really surprising assuming that the QLL may be only a few layers of water molecules?**

We agree and have removed the phrase "somewhat surprisingly"

**I recommend include at least once the correct technical terminology for the compound "Rose Bengal" in the manuscript.**

We have added the correct chemical name for Rose Bengal.

**Comments from Anonymous Referee # 2**

**I found this paper interesting and easy to read, and agree with the authors that it is important that the effect of sample preparation method is considered when interpreting the results of analyses on laboratory samples. This article is relevant to The Cryosphere and its readership, demonstrating the results of a laboratory study on experimentally grown ice.**

**I would recommend that this article be published, after minor revision. I found several areas where I would have appreciated more information and/or discussion on both the choice of sample preparation methods, and the discussion of results. I have split my recommendations into General comments/Specific comments and Typographical comments.**

**General comments**

**Introduction/Methods**
**From the introduction as it is written, it is unclear to me why these three sample preparation methods have been chosen. The authors suggest that their results are relevant to solute positions in snowpack, but the sample preparation procedures here - freezing directly from a liquid solution to a (presumably crystalline) solid – do not seem to be immediately comparable with snowpack formation. Additional discussion of snowpack formation mechanisms, and/or justification of these sample preparation methods, would improve the paper, clarifying the reasoning behind these experiments. (Have these sample preparation methods been used in previous experimental studies? Does the resultant structure represent snowpack well?)**

We have revised both the introduction and methods sections to explain the choice of freezing methods, and to refer to text in the results section discussing expected freezing behavior.

**Why were CsCl and Rose Bengal chosen? Are these solutes present in real snowpack? If not, what is the reasoning behind using them, are they expected to behave like the real solutes? (And what is the evidence (with references) for this?)**

CsCl was chosen because of its high aqueous solubility (suggesting all the material will be present in LLRs, rather than precipitated into the ice matrix) and large Xray absorption cross section (enabling ready detection and quantitation in the microCT).  Rose Bengal was chosen for similar reasons.  While these solutes are not commonly found in snowpacks, we chose them because they represent surrogates for some common compounds in snowpacks, such as sodium chloride (with CsCl as a surrogate) and polycyclic aromatic hydrocarbons (with Rose Bengal as a surrogate).  Clearly, two model compounds cannot reflect the diversity of materials found in natural snowpacks.  However, based on the structure of the materials, we believe they will behave in ways similar to other solutes in snow, and can be used to guide further research.

**The samples frozen are of different sizes (Table 1). Is it possible to compare samples of different volume? Will the greater volume of the Freezer and LN2 samples not also affect the freezing behaviour?**

Much of the behavior we have noted in our work, such as the association of solutes and air bubbles, seems to be independent of volume.  While it is possible the greater volume of the Freezer and LN2 samples may have some minor effect on freezing behavior, we have seen no evidence to suggest sample volume would significantly affect sample morphology.  In some cases, we conducted identical experiments in two different volumes, and saw similar patterns in freezing morphology independent of sample volume.

**If I understand correctly, you make the assumption with your calibration curve that the effect of CsCl on the radiodensity of water and ice is the same. And so all results based on this method are dependent on this assumption. What is the support behind this assumption? (Page 7).**

Measured radiodensity is a function of the physical density of the material (as shown by the comparison of ice and water radiodensity on page 7) and the electron density of the atoms present in the material. Because electron density is an atomic-scale property, we believe CsCl electron density would remain the same in both ice and water. In the case of the physical density of either the CsCl-water or CsCl-ice mixtures, it is possible that the presence of CsCl could affect the density of these preparations differently. In order for such a difference to be noticeable in our work, however, the density change would have to be significant in both ice and water, and in the opposite direction (for example, with the addition of solute to ice making ice more dense, while solute-containing water were less dense). In addition, because of the low concentrations of solute in our experiments, we do not believe any density differences will have a noticeable impact on our results.

**Discussion**
**Were there any measurements of the grain size/shape of the ice in these samples? And the position of the pores/solutes with respect to grain boundaries? Might the grain size (likely controlled by the freezing temperature and subsequent annealing) play a role in the position/size of solute pockets?**

We did not attempt to measure grain size or shape, nor the location of pores or solutes relative to grain boundaries. We do believe grain size (and morphology) could play a role in the position and size of solute pockets. Unfortunately, we do not currently have the equipment to identify and study individual grains; we hope to address this question in the future.

**The experiments are carried out with a high concentration of solutes (with respect to inland snows). Additional discussion as to the impact of this higher concentration on the results would be enlightening – will a lower concentration have similar results?**

We have not yet investigated whether lower concentrations of solute will alter the freezing pattern. We agree such work would be enlightening, however. We chose our test concentration (1 mM) as a compromise between solute concentrations in natural snowpacks and the need to have enough solute present for easy visualization.

**I find the result that solutes are often associated with air bubbles in the Freezer samples, but less so for Freeze Chamber samples, very interesting – do you have any ideas of potential mechanisms which would be working differently in the two situations?**

We also find this an interesting observation. We do not have a clear idea why this is the case, but suspect it is related to freezing speed. In the Freezer samples, the ice matrix may form slowly enough to allow the ice matrix to exclude both gases and solute to the same compartments. In the more quickly frozen Freeze Chamber samples, the ice matrix may form in such a way to allow smaller inclusions, which may preferentially form for either solute or gas, but not for both.

**Specific comments**
**Page 3 Line 27. Define PAH at the first usage.**

Corrected.

**Page 4 Line 4. "the cage effect at a given temperature". It is unclear what the cage effect is, an extra sentence explaining this would be helpful.**

We have added an explanation of the cage effect.

**Line 9. Not clear what is meant by "compartments"**

We have changed the wording to use the term "reservoir", used elsewhere in the paper.

**Page 5 Line 4. You have given example of qualitative results, (visual), is there an equivalent example for quantitative?**

We have clarified that our quantitative results are given in both tabular and graphical formats.

**Line 17. What is the (potential) effect of the elevated concentration of solutes?**

It is difficult for us to assess the impact of the solute concentration. Our best guess is the morphology would generally be similar at different concentrations, but the number of inclusions would vary. Figure 2 provides some clues; when the amount of dissolved gas present was reduced, the size of the bubbles was the same, but there were fewer bubbles. Further research would help address the question of how solute concentration would affect sample morphology.

**Page 6 Line 4 Did you consider the effect of the annealing time? "At least 1 hour" – were some samples annealed for much longer? What effect might this have/did this have on the solute/pore distribution?**

Some samples were annealed for as many as four hours. We do not think annealing time would have a significant effect on our sample morphology. We annealed the samples to give them time to transition from amorphous ice, to cubic ice, and finally to hexagonal ice (Beine and Anastasio, 2011). At -10 °C, the speed of this transition should be on the order of minutes (Hobbs, 1974), so our minimum 1 hour annealing time should be more than adequate. Once the ice has transitioned to hexagonal ice, we do not expect any further changes to the matrix.

**Line 5 - You specify the thickness of polypropylene vial walls, you should also specify the glass wall thickness. Is this the same thickness? If it isn't, any thermal consideration as to the differences between the samples frozen in glass and plastic is not only due to the difference in thermal conductivity between glass and plastic, but also the thickness. Any difference between glass and plastic vials (even though there likely will be) cannot be evidenced by these experiments if the thickness is not the same.**

We have added the glass vial wall thickness (0.8 mm). While we agree the wall thickness is a contributing factor in how the vials will transmit heat and therefore may affect how the sample will freeze, we believe the small difference in wall thickness between glass and plastic vials (approximately 20%) compared to the large difference in thermal conductivity between the two materials suggests material choice will be a more significant factor.

**Line 28 "small amounts of sample" – can you quantify this?**

As the sample container is not a perfect geometric shape, we have to use radiodensity differences to determine where the sample ends and where the vial begins. Because segmentation requires analysis of very large numbers of voxels, we have developed a semi-automatic process to do this, rather than individually inspecting each voxel. Therefore, we do not have a reliable way to quantify the impact of this process. We estimate the amount of sample excluded here would be well under 1% of the overall sample volume.

**Line 29 "very little sample in contact with air.." – similarly, is this quantifiable?**

Similarly to the response directly above, we believe the amount of air voxels incorrectly included in the sample is well under 1% of the overall sample volume.

**Page 7 Line 25 – the process of solute rejection on freezing has been seen in several systems, including papers earlier than the referenced Cho paper. I would suggest further literature should be included here (and possibly in other places) to avoid overreliance on this one article.**

**E.g. starting points for the sea ice system: Lake and Lewis 1970 J Geophys Res 75 3; Wettlaufer et al J. Fluid Mech. (1997), vol. 344, pp. 291-316. Other systems (metals): Worster J Fluid Mech. (1992), vol. 237, p p . 649-669**

We have added additional literature as suggested, and thank the reviewer for these references.

**Page 8 Line 2 – what is the temperature effect on the solubility of CsCl? I would anticipate the solubility to decrease with a decrease in temperature – so the value you quote here at 20C would not be appropriate at -10C. A difference in solubility will change the calculated masses.**

**A quick literature search provided me with this, but there are probably other options in the literature: Jiang et al, 2003 Indian Journal of Chemical Technology Vol. 10, 391-395**

The solubility limit of CsCl does decrease with a decrease in temperature, and is 9.6 M at 0° C (NIH, 2015); we have added this value to the paper. Jiang et al. (Jiang et al., 2003) shows the decrease in solubility with temperature is roughly linear from 20° C to -10° C, so we expect CsCl solubility would still be well above the expected concentration in the LLR of 2.7 M. Since masses were calculated based on the assumption of CsCl present in LLRs with a concentration of 2.7 M (well below the expected solubility limit for CsCl in the system), we do not believe the calculated masses would change.

**Line 5 – are there no other effects on the radiodensity than the concentration of solute? What about temperature?**

Radiodensity is a function of the electron density of the imaged material. So, the concentration of the solute and the background radiodensity of the solvent are the only significant factors in determining the radiodensity of the sample. Temperature can have a small effect, in that sample density, and therefore effective concentration, does depend on temperature. However, within the temperature range of our experiments, the impact of temperature on density changes of either water or ice is minimal.

**Line 14 – reminder of the value used for pure ice radiodensity would be helpful here.**

We have added this value (4948).

**Line 18-21 I find this sentence complicated to understand – is there a better way of presenting the four domains – in particular the distinction between 2-10% and 10% LLRs is not clear to me without rereading several times.**

We have rewritten the sentence to make it clearer.

**Page 9 Line 2 – can you put a number on the amount of solute "lost" in this process?  Quantitative idea of the effect that the threshold has?**

We do include the solute present in the matrix we have labeled "water ice" in Table 1, and discuss solute in this compartment in the text.  We have calculated the mass of solute present in the water ice compartment by subtracting the total mass in the sample from the mass in the other more concentrated LLR compartments.

We initially attempted to determine the mass in the water ice compartment directly, by separating the pure ice voxels from the voxels containing <2% LLRs.  This proved to be statistically challenging and gave poor results, so we did not pursue this effort further.  We have added additional explanation of this point in the text.

**Line 4 – why only carry out this calculation for some samples? Why not all?**

We have reworded this sentence to clarify – we only calculated CsCl mass for CsCl-containing samples, not gas-only samples or Rose Bengal samples.

**Line 14-16 – The different thermal conductivities between water and glass has no impact here – even if you had two materials with the same thermal conductivities, the sample would still freeze from the outside in. Suggest suppressing the reference to thermal conductivity.**

Agreed, we have eliminated the reference to thermal conductivity.

**Suppl Fig 1 - The directionality of bubbles in your Supplementary Figure 1 also seems to support freezing from the exterior of the sample – e.g. Carte 1961 talks about direction of bubble formation in a temperature gradient (Proc Phys Soc 77, http://iopscience.iop.org/article/10.1088/0370-1328/77/3/327/pdf.), I'm sure there are other references as well.**

We thank the reviewer for the reference, and have modified the text to include this reference and further discussion of the bubble shape and morphology.

**Line 21 can you quantify the size of the bubbles in the two figures?**

We have quantified the approximate bubble size and added this information to the text.

**Fig 2c – adding a vertical line and label at the central point of the air peak would make the histogram easier to interpret, with an equivalent label for the pure ice peak at VLLR/VVOXEL = 0.**

We have added arrows to indicate the air and ice peaks, which seemed to be a bit clearer than adding additional vertical lines, but accomplishes the same goal.

**Page 10 Line 2 – this would read better if the sentence was inverted to mention log scale before the "clearly..." comment. E.g. "Taking into account the log scale... the volume is clearly less..."**

Agreed, we have made the suggested change.

**Line 12 – What are you basing the expectations of freezing speed/direction of heat removal/position of solute inclusions on? Need a reference (or more explanation of the reasoning).**

We have added additional description of our reasoning.

**Line 21 – can you find a more technical term than "blow up"? –e.g. magnification/detail...**

We have changed "blowup" to magnification, on Line 21 and in three other places.

**Line 22 – do you have a mechanism/an idea for the link between solute pocket position and bubbles?**

We speculate that the colocation is related to exclusion of gases and solutes by the forming ice matrix. The freezing front will tend to push the gases and LLRs ahead of it, with the LLR likely remaining at the freezing front and the gas furthest from the ice. As multiple freezing fronts meet, the gas bubbles will meet and enlarge, with the areas of concentrated solute encircling the bubbles.

**Line 24 – The use of the word "identically" here is disingenuous, the samples are not produced identically (I agree the solutions may have been). But as the aim of the paper is to demonstrate the differences when samples are produced using nonidentical ls, it would bring this message home more convincingly if you avoid referring to differently frozen samples as "identically" produced. (There is another place this happens later on as well – Page 14, Line 25).**

We agree the use of "identically" here can be confusing, and have changed the term to "similarly" to avoid confusion. We made the same change for Page 14, Line 25.

**Line 29 – "blow up" – as above.**

Replaced with "magnification".

**Page 11 Line 1 – "surprisingly different" – are these morphologies repeatable? To me it is not that "surprising" that a difference in freezing front gives a difference in solute distribution – maybe a different word than "surprisingly" could be used – I agree that it is interesting that this happens and that you have been able to observe it. Also, you are comparing samples of different size, this will cool/freeze differently regardless of environment.**

The morphologies were repeatable. We have replaced "surprisingly" with "substantially".

**Line 10 – Why would a freezing front process only affect the solutes and not air bubbles? I would be interested in an expansion on a theory for the mechanism for this.**

We are not sure why a freezing front would affect only solutes and not air bubbles, and do not have a working theory to explain the observations presented here. Nonetheless, we do see segregation of solute towards the surface of the sample.

**Line 23 – "No air bubbles..." I don't agree there is nothing in FigS6 – I see one inclusion/something in Fig S6.**

We believe that single spot is a deformity in the wall of the sample container, which subsequently shows up on the video. We have added additional explanation to the figure caption. Compared to the results obtained using the previous two freezing methods, notable features are absent in the LN2 sample.

**Fig 3g – (g) label appears twice. Labelling of individual curves is unclear.**

We have removed the duplicate figure letter. Line identity is indicated both on the figure with labels next to the lines, and in the figure caption by reference to the line color. Which aspect of line labels remains unclear?

**Line 28 – how does histogram show that voxels contain concentrated solutes?**

In the inset graph, the orange line (LN2 sample) has some voxels at the right end of the curve with $V_{LLR}/V_{Voxel}$ greater than that in the pure water sample. The histogram indicates a higher radiodensity in these voxels, which in turn shows the presence of solute. Because solutes will be present in LLRs, these voxels contain concentrated solutes. We have changed the test to clarify this point.

**Page 12 Line 5 – If the effect is barely visible, does that necessarily mean it is not there/definitely not important?**

No, in fact we believe this is an important finding – even rapid freezing in liquid nitrogen can likely result in concentrated LLRs. We have revised the text to avoid diminishing this finding.

**Line 12 – where are the concentrated solutes in Figure 3f? It is not clear in either Fig 3f or Fig S9 where these are – there seem to be only air bubbles.**

Some small inclusions visible in other data (not presented) were smoothed away during image processing, but the text was not changed to reflect this. We have revised the text to match the figures.

**Line 17-19 – this sentence is poorly worded ("much different than ...") . Can you expand on the reasons why the ice matrix would be modified? There needs to be a reference for this.**

We agree the phrase beginning with "much different than…." is awkward and removed it. As we state in the text, the reasons behind the change in morphology is not clear to us, and we do not have a strong hypothesis for why the ice matrix would be modified. Therefore do not have any references to support this idea. We did, however, feel it appropriate to present two possible ideas as to why the ice matrix would form differently in the presence of different solutes.

**Line 24 – precipitates in LLRs? – but these LLRs are not visible in Figure 3f or Supplementary Figure 6? So how could there be precipitates within them?**

As discussed previously in the text (page 7 lines 1-3), mathematical smoothing can eliminate small (~80 μm in diameter) features, such as LLRs. While the LLRs are not visible in Figure 3f or Supplemental Figure 7, we believe they are present. The LN2 results in Figure 3e show no inclusions, while a higher-resolution version of that figure (Supplemental Figure 7) reveals small bubbles and solute inclusions. It is possible Rose Bengal could be present as a precipitate in small inclusions, yet not visible in our reconstructed images.

**Line 25 – it would improve the flow of this section if the discussion about the histogram was combined with earlier discussion (Page 11, line 7-8) as the earlier part seems truncated and unfinished.**

We agree and have made the suggested change.

**Page 13 Line 22/Fig S10 – do you only have replicates for the freeze chamber? It would be interesting to have the equivalent histograms for each method.**

Creating the histograms requires segmenting and then analyzing the imaging data, which is a time-consuming process. Therefore, we only created the comparison histograms for these three freeze chamber samples. Visual inspection of results for the other sample types (Freezer and LN2) suggests the differences between freezing methods are reproducible.

**Line 24 – You cannot say that the "two variables [freezing method and solute] are the primary factors influencing ice morphology" as you only change these variables in your experiments, so of course they are the two primary influencing factors here.**

We have revised the text to remove this implication these are the only factors at work in the system.

**Page 14 Line 1 – what are the errors on the VLLR/VVOXEL = 2-10% for the two methods? These would be useful to determine whether the factor of 2 difference is reliable – as a difference between 0.003 and 0.006 doesn't seem large (but may still be significant).**

Because of the effort required to determine $V_{LLR}/V_{Voxel}$ for each sample replicate, we do not have enough replicates to adequately characterize the error of $V_{LLR}/V_{Voxel}$.

**Line 24 – If I understand correctly, LLRs are present when there is solute extruded from the crystallising ice – so why do you have LLRs in this (pure ice) sample?**

The legend on this figure mistakenly and confusingly includes reference to LLRs. We have removed that legend line.

**Line 25 – use of "identical" confusing again.**

We have changed the word to "similar".

**Line 29 – "while the reason for this morphology is unclear..." the morphology of bubbles, and the effect of a temperature gradient, has been studied previously – e.g. Proceedings of the Physical Society, Volume 77, Number 3 http://iopscience.iop.org/article/10.1088/0370-1328/77/3/327/pdf.**

We thank the reviewer for the reference, and have changed the text accordingly.

**Page 15 Line 15- Where are the "elongated solute inclusions" in Figure S14. They seem no more or less elongated than the air bubbles.**

We have revised this sentence and eliminated reference to elongated solute inclusions.

**Figure 5 – is this a vertical slice? Specify this in the caption. Also - "along the direction of the temperature gradient" – specify which direction – i.e. from colder to warmer.**

We have clarified that the images are vertical slices and specified the direction of the temperature gradient.

**Page 16 Line 16 – a reference for "melting into the bulk ice" is required (e.g. Movement of brine pockets by salt diffusion - Notz D and Worster MG (2009) Desalination processes of sea ice revisited. J. Geophys. Res. Ocean., 114(5), C05006 (doi:10.1029/2008JC004885))**

We have added the reference as recommended.

**Line 20 – what is the concentration of solutes in clean polar samples? This should be mentioned in the introduction as well.**

Of course the concentration can vary considerably, but concentration of solutes in clean polar snow is on the order of 10 μM (Yang et al., 1996). We have added this information to the introduction.

**Line 21 – "significant impacts" - I think significant should only be used in statistical context, so here "important" or similar would be more appropriate. There should also be more discussion on how this assumption that a lower concentration will have the same effect was made.**

We have changed the word "significant" to "important", and revised the sentence to state that we believe lower solute concentrations may also impact sample morphology.

**Page 17 Line 3 – this sentence ("Cesium chloride...") starts abruptly, would be improved with an introductory couple of words.**

We have added several words to this sentence to bridge from the previous paragraph.

**Line 14-15 "Our work here can help guide further investigations to understand the driving forces shaping snow and ice structures in the natural world, as well as the rate of chemical reactions in snow and ice." – it is not clear to me how this work helps to understand the driving forces etc, nor how there is any (direct) impact on studies into the rate of chemical reactions.**

We believe our findings suggest both freezing method and the nature of the solutes present are important to determining the morphology of ice matrices, both natural and artificial. For studying the rate of chemical reactions, the location of solute in laboratory samples may have significant impacts on the measured reaction rates, so knowing how freezing method can influence the solute location can directly impact these studies. We have revised the text to address the reviewer's suggestion.

**Line 16-19 "At the same time, our results suggest subtle changes in the preparation of laboratory ice samples can have significant impacts on the location of solutes in samples, requiring careful and consistent sample preparation to ensure meaningful results." - this part is the important conclusion from this work. I would therefore suggest removing/reducing/rephrasing the first half of this paragraph, and expanding on the second half.**

We have rearranged the paragraph as suggested, and rewritten parts of it to incorporate the reviewer's suggestion.

**Supplementary info**
**It would be more coherent in there was only one format for videos and one for images in the supplementary information.**

We agree this would be desirable and more coherent. However, because of the various software packages used to generate the figures and videos in the supplemental material, the initial output formats will necessarily differ. While it is possible to standardize the format, we have used common formats available to users of standard computers, and therefore don't believe the end users have been inconvenienced.

**It would be useful to have brief captions/explanations of supplementary images and videos in the supplementary information, to avoid having to search through the main text for the information.**

We appreciate the suggestion and have added a caption file describing each supplementary image or movie.

**The colours marked "orange" in the text do not appear orange in the videos (at least on my screens), they are yellow.**

We do agree the color can appear yellow on some monitors. Since the only other color in the videos is red, which seems to appear as red on all systems, the remaining color should not confuse the reader, even if it appears more yellowish than orange. We have indicated this caution in the captions file, and we have changed the legend in the videos to call the color yellow rather than orange.

**Typographical errors**
**Pg 3 Line 6 "to form hydroxyl radicals" Pg 5 Line7 – missing word? "was produced from..." Pg 5 Line 10 - repetition of "its" Pg 8 Line 17 "VVOXEL" should be in capitals Pg 11 (22-23) "Results for a 1.0mM CsCl sample.. are shown in Fig. 3e"**

We have corrected these errors.

**Figure 4 – the blue arrows are not visible when printed in black and white.**

We see the arrows when printed in black and white, and appreciate the reviewer's observation. We have changed the arrows to yellow to provide better contrast.

**Comments from S. Maus (Referee)**

**The paper presents and discusses results from X-ray tomographic imaging of aqueous solutions frozen in small containers in the laboratory. With two solutions, cesium chloride (CsCl) and Rose Bengal solution, three different freezing methods were used: (i) freezing by putting containers in a normal freezer, (ii) unidirectional bottom-up freezing with containers placed on a cold plate and (ii) putting small vials into liquid nitrogen. The frozen samples are imaged my X-ray microtomography, mostly at a voxel size of 16 $\mu$m, to obtain 3-d greyscale images of X-ray transmission. After segmentating the greyscale images into different classes that reflect solute and air or gas content, the authors discuss these images qualitatively in terms of distribution of solute inclusions and air bubbles. The authors also perform a quantitative analysis of the distribution and content of solute in liquid like regions (LLRs) versus solute incorporated in the solid ice matrix, as well as some observations on the movement of liquid inclusions. The authors conclude that the work shows that *the structure of laboratory ice samples, including the location of solutes, is sensitive to freezing method, sample container, and solute characteristics, requiring careful experimental design and interpretation of results*. The work is proposed to enhance our understanding of solute segregation in natural snow and ice, as well as of *the air-ice interface and liquid-like regions within the ice matrix*.**

**I agree with two other referees that the paper is relevant for ˊThe Cryosphere' and its readership, and also mostly with their comments. I would like to add comments on two major issues on which the paper in my opinion needs improvement. First, I encourage the authors to improve review and referencing of the published background on ice observations and solute redistribution during freezing, to be included in the introduction and discussion of their own observations. Second, I think that there is potential for improvement in the quantitative analysis of the 3-d images, and a**

**critical discussion of the method and results. In my comments I will give literature examples that I hope will help the authors to improve their analysis and presentation.**

**Main concerns I-III**

**I. Background - ice, freezing and solute inclusions**

**1. In the literature on ice physics and chemistry there are several books that include fundamental discussions of ice structure and solute distribution during freezing but none of these is mentioned. I recommend to have a look into the literature (e.g., Shumskii, 1955; Hobbs, 1974; Lock, 1990; Petrenko and Whitworth, 1999; Pruppacher and Klett, 1997), and possibly cite from there.**

We thank the reviewer for these references, and have included them as appropriate.

**2. It is well known that during the freezing of saline solutions most solute is rejected into the remaining mother solution and not incorporated into the solid ice matrix (e.g., Hobbs, 1974; Petrenko and Whitworth, 1999). This fact should be more clearly mentioned in the text. The authors for example write (P16, L13-16)** *While the air bubbles remain stationary in the ice matrix, the CsCl moves, consistent with the idea that solutes are present as a concentrated liquid-like solution, which can migrate either along the boundaries between air bubbles and the bulk ice, or possibly by melting into the bulk ice itself.* **Such formulation indicates that solute rejection during solidification of water is only an** *idea* **rather than a fact under most conditions.**

We have revised the noted section, as well as the introduction and abstract, to state the exclusion as a well-established finding rather than an idea.

**3. I am also missing background literature on observations of solute inclusions in ice. For example, already Quincke (1905) has described the morphology and distribution of liquid inclusions of ice grown from saline solutions, and there is much more information in the books on ice physics mentioned above. As an example, Shumskii (1955) describes the solute distribution in ice as (p.180): 'The distance between neighbouring interlayers of inclusions in a crystal decreases with increasing concentration of impurities in the remainder mother solution; often this distance is as much as 35-45 $\mu$ with inclusions 8-15 $\mu$ thick'. Such information is certainly relevant for the discussion and interpretation of the results in the present paper.**

We have included reference to previous work on solute inclusions, and tried to use them to guide interpretation of our results.

**4. An idea of the expected microstructure and potential separation of solute inclusions and air bubbles may be obtained by consulting published work based on thin section analysis of frozen solution or pure water droplets (e.g., Hallett, 1964; Rohatgi and Adams, 1967). For example, a useful information from these studies is the dependence of dendrite or plate spacing on freezing velocity: The faster the freezing, the smaller the spacing of ice plates and solute inclusions, which normally implies smaller dimensions of solute inclusions. For the present study this may affect the detectability of solutes, especially for the samples frozen very rapidly in liquid nitrogen-based freezing.**

We thank the reviewer for these references, and have incorporated them into our text.

**5. Earlier basic work on solubility of ions in the solid matrix as well as solute partitioning at a freezing interface (e.g., Tiller, 1963; Gross et al., 1975, 1977) should be mentioned and discussed.**

**Such information is particularly important when it comes to the quantitative analysis and discussion - see my comments below.**

We have incorporated ion solubility in the solid matrix as appropriate.

**6. How is the freezing point depression the authors assume for CsCl (2.7 M CsCl at -10 ℃) computed, or on which reference is it based? E.g. according to Pruppacher and Klett (1997) (p. 125, Fig. 4-12) one might expect that a value of 3.1 to 3.3 M is a more realistic value at -10 ℃. While such a change in equilibrium concentration would affect the estimates of the volume of LLR from the greyscale images, it would not affect the solute content within liquid inclusions. However, it would decrease the maximum solute content in voxels in the histogram, and thus may give hints on the proper estimation and possible rescaling of equation (1).**

Our use of 2.7 M is based on theoretical thermodynamic calculations, and corresponds to the ideal case given in Pruppacher and Klett (Pruppacher and Klett, 2010) Figure 4-12. Pruppacher and Klett and Haynes (Haynes, 2014) both present data for the freezing point depression of CsCl, but only up to a salt concentration of 1.8 M (Pruppacher and Klett) or 1.4 M (Haynes). Extrapolating their data to the concentrations expected in our samples (i.e., at -10 °C) suggest the CsCl concentration in LLRs would be somewhere between 3 and 3.2 M, i.e., 10 – 20% higher than our ideal case concentration, but neither source presents freezing point depression data measured at such a high concentration. In the absence of measured information for the actual composition of CsCl solutions under our experimental conditions, we have elected to stay with the theoretical prediction of salt concentration of 2.7 M, especially since the change to non-ideal conditions would be relatively small (i.e., 20% or less). However, we have altered the text to note the uncertainties in the CsCl concentration in LLRs at -10 °C and explain the potential impact of the non-ideal concentration on our findings.

**7. X-ray tomography of solutions frozen in the laboratory has been performed earlier Miedaner (2007); Miedaner et al. (2007) and may be compared to the present results.**

We have included these references both in the introduction and in the text as appropriate.

**II. Image segmentation and quantitative analysis of solute content and locations**
**1. The proposed segmentation approach is based on equation (1) on page 8 assuming 2.7 M equilibrium concentration CsCl at -10 ℃. Please have a closer look in the literature to evaluate the uncertainty of this estimate.**

We have done so and expanded the text to better describe uncertainties of our estimate, as discussed in the response to point I.6. above.

**2. The choice of a cutoff at *LLR* = 2% to estimate the solute content in liquid like regions seems somewhat arbitrary. As the pure ice histogram in Fig. 2c extends to slightly above *LLR* = 4%, rather such a cutoff would be more consistent. At least would a *LLR* = 4% cutoff define a lower bound of the solute content in liquid inclusions.**

We agree, the choice of cutoff is somewhat arbitrary, and that a cutoff of 4% would virtually eliminate the chance that a voxel with LLR > 4% actually is pure ice. However, such a choice could also incorrectly imply that many voxels with an LLR < 4% contains no solute. As stated on Page 8, Line 25ff, we chose 2% as the cutoff because it is three standard deviations above the mean for LLR in pure ice voxels. Therefore, the number of voxels that are actually pure ice but are mistakenly included as containing solute is quite small, and this value seemed a reasonable cutoff for voxels containing appreciable solute. Note, also, the graph may be visually misleading, as the Y axis is a log scale, imply a larger number of voxels above 2% than are actually present.

**3. As the authors correctly point out, the solute content will be underestimated due to the possibility of mixed air and solute pixels, which may then have radio densities between brine and air, and thus be classified as ice. Can this bias be estimated? An approach to place a bound on this bias could be to count the surface voxels of the air bubbles and assume that these contain CsCl brine and air. How would this affect the results in Table 1?**

We considered various methods to address this issue quantitatively, but found no practical solution. Given the large number of voxels present, manual counting was not possible, and the software tools available to us did not have the capabilities to perform this analysis. We agree it represents an uncertainty in our analysis, and have added additional text to discuss the problem.

**4. In the histogram in Fig. 3 the maximum volume fraction of liquid is roughly 0.9. However, if calibration and equation (1) would be correct I would expect that the maximum value should at least be 1 (due to expected noise even larger), provided that there are at least some liquid inclusions that exceed the volume of a voxel with side length 16$\mu$m. While this may not be the case, I would expect that it would very likely be the case for high resolution imaging with 2$\mu$m voxel size. How does such a histogram look like, and may it be used to improve the calibration in eq. (1)?**

We agree such an analysis would be useful and interesting and useful. Unfortunately, to do such an analysis would require imaging aqueous solutions at 2 µm resolution to create a standard curve, as was done at the 16 µm resolution samples. We did not have the resources to do such a standard curve, although we hope to further investigate higher-resolution imaging in the future.

**5. According to Table 1 the solute content classified in solute inclusions is 12-35 %, and the remainder is concluded to be incorporated in the ice matrix. How does this compare to expected solubility limits in the solid? E.g., Gross et al. (1975) suggested a solubility limit of 1-2 ×10−4 M for HCl in the solid ice matrix. The result from the authors calculations (65 to 88% of the initial 1 mM CsCl in the ice matrix) would roughly imply a 3-9 times larger solubility of CsCl in ice. Do any studies exist that support such a high solubilty of CsCl in solid ice? If not, then this might be another indication that eq. (1) should be changed by a prefactor that gives larger liquid fractions of at least 1 at the higher end of the histogram. Again, it appears very important to present a similar analysis of high resolution images, that could solve this problem.**

We agree with the estimate of 65% of CsCl present in "water ice", as given in Table 1. However, we do not believe all the CsCl present in this water is present as solute in the solid ice matrix. Rather, most of this solute is likely present as LLRs where $V_{LLR}/V_{VOXEL}$ is < 2%. Due to resolution and background noise, our imaging system cannot reliably distinguish between voxels where, for example, $V_{LLR}/V_{VOXEL}$ = 1%, and a voxel containing only pure water ice. However, higher resolution imaging results, such as Figure 4 or Supplemental Figure 7, suggests smaller inclusions are indeed present.

The "missing" CsCl mass here is 0.65 * 126.3 µg = 82.1 µg, or 0.49 µmol. Assuming this solute is entirely present as LLRs with solute concentration of 2.7 M, this equates to a total LLR volume of 0.18 µL. The volume of pure ice (again from Table 1) is 716 µL. Therefore, assuming the remaining CsCl is distributed equally throughout the voxels labeled as pure ice in Table 1, the calculated average $V_{LLR}/V_{VOXEL}$ for these voxels is 0.025%, indistinguishable from water ice in our system. While it is possible the CsCl is present (at least partially) as solutes in the solid ice matrix, we believe it is more likely to be present primarily as small LLR inclusions.

We have revised the text, incorporating also the ideas from I. 6. above, to more clearly address this issue.

**6. The inset in Figure 3 compares the histogram envelope around the radiodensity of ice for the Milli-Q and solute samples. It indicates that the ice peak and envelope in the histogram is slightly shifted to the right for the frozen solutions with respect to frozen Milli-Q - which is particularly apparent for the LN2 samples. Such a shift would indeed be consistent with Cs and Cl incorporated in the solid ice matrix, where they act in the same way as strong X-ray absorbers as when in liquid solution. It would be very interesting to evaluate, if it is possible to estimate the solute content in the ice matrix from this shift.**

We also noted that shift and agree it is an interesting finding. We agree it probably represents Cs and Cl incorporated into the ice matrix. We did attempt to estimate the solute content by subtracting the histogram of the solute-containing sample from a sample of pure MilliQ water. Unfortunately, this process was very sensitive to variability in the distributions of the background samples, and we were not successful. To understand this variability, we would have had to conduct a number of imaging studies on various days of both pure water ice and CsCl solutions, an effort we unfortunately did not have the resources to conduct. We do think it is a worthwhile activity, and hope to do this in the future.

**7. A statistical analysis of size distribution of solute inclusions and air bubbles would be very helpful. Such a statistics would also justify to include the results from Rose Bengal solutions, that else is given too little weight in this study.**

We agree that such an analysis would be worthwhile and useful. Such an analysis, however, would require considerable time and resources, and we were unable to complete it for this paper. We hope to revisit this issue in the future.

**III. Results and Discussion**
**1. Every paragraph in the discussion contains a reference to supplementary material, that is the discussion is based very much on the latter (S1-S16). While it is helpful to provide such material, I regard it as inappropriate to build up the discussion of a research paper on that much supplementary information. Some of this information should become part of the paper and the discussion should be rewritten.**

We understand and share the reviewer's concern, but are somewhat restricted by the image formats available to us in a standard scientific paper. Of the 16 Supplemental files, all but 3 are video files. Of the video files, most are represented as individual images in the paper itself. It is not possible to incorporate these videos into the paper as currently structured for The Cryosphere.

**Specific comments**
**P 4, L 23-25 –>** *But to our knowledge this method has not been used to investigate the structure and solute locations for laboratory samples prepared under controlled conditions with specific solutes* **- I would not call the freezing conditions** *controlled***, as neither cooling rates or supercooling in the samples were controlled or measured.**

While we do no present measurements of cooling rates, the sample preparation methods were controlled in the sense that the freezing methods were repeatable. We have changed the word "controlled" to "reproducible", to avoid implying we cooled the sample at a standard rate.

**P 4, L 29 –>** *In this work we focus on cesium chloride (CsCl) as our solute. However, because a previous study (Cheng et al., 2010) found different solutes can affect freezing morphology and therefore may influence solute location, we also imaged ice containing the organic compound Rose Bengal.* **I suppose that CsCl was chosen because it warrants a high X-ray contrast between ice and**

solute. **Why was Rose Bengal chosen? Also, as the results presented are, except for a histogram in Fig. 3 as well as supplementary material, for the CsCl solutions, I would rather suggest to remove the few Rose Bengal results and notes, and rather present a systematic and quantitative comparison elsewhere.**

In addition to its high Xray absorption cross section as noted by the reviewer, CsCl was also selected because of its high aqueous solubility. Rose Bengal was chosen because of its relatively high aqueous solubility (for a chemical of its size) and large Xray absorption cross section. While we agree experimental results for CsCl are far more extensive than for Rose Bengal, we do think including this compound is essential to demonstrating the idea that solute choice can make a significant difference in sample morphology. We hope to present a more extensive and quantitative evaluation of solute effects in the future.

**P 5 L 1 –>** *Cheng et al., 2010* **- this is a reference to a study based on a rather different method, that yields the surface distribution of solutes/ions. There exist other studies that have shown the influence of solute on freezing pattern, for example the mentioned work by Rohatgi and Adams (1967). I cannot see that the cited paper is an argument to use Rose Bengal as an alternative solution.**

We offer the cited paper not as an argument to use Rose Bengal in particular, but as a motivation to study solutes other than CsCl. We have added the Rohatgi and Adams reference to this sentence.

**P 16 L 13 –>** *While the air bubbles remain stationary in the ice matrix, the CsCl moves, consistent with the idea that solutes are present as a concentrated liquid-like solution, which can migrate either along the boundaries between air bubbles and the bulk ice, or possibly by melting into the bulk ice itself* **- First, I find it surprising, that the air bubbles remain stationary, because it is well established that air bubbles migrate in a temperature gradient at similar rates as liquid inclusions (Dadic et al., 2010). Second, some reference on the process of brine pocket migration should be mentioned here, please have a look at Light et al. (2009) and the literature reviewed therein. Third, Light et al. (2009) also found migration for solid crystals, so the movement of solute is no proof for its liquid character.**

"Stationary" here may be a relative term; we did not note migration of the large air bubbles in the sample shown, but we did not attempt to rigorously quantify this motion, either. We do note that Dadic et al. (2010) presents measured data of bubble migration in laboratory samples, and found rates at -10 C ranging from 1.5-3 $\mu$m h$^{-1}$/(K$^{-1}$ cm$^{-1}$). Brine inclusions (original data from Light et al. 2009, and also cited in Figure 7 in Dadic et al. 2010), however, moved at approximately 10 $\mu$m h$^{-1}$/(K$^{-1}$ cm$^{-1}$). While we agree with the reviewer these rates are roughly similar, these findings also suggest bubbles and brine inclusions may move at different rates, and are consistent with our observation that solute inclusions move faster than air bubbles. Additionally, we note that our system may not be accurately modeled by comparison to pure brine inclusions or air bubbles. Light et al. (2009) notes that "dissolved gases may play a role in the migration of brine inclusions" and further states that "The effect of included gas bubbles on brine migration has not been studied."

We have revised the text to include these ideas and the other suggestions made by the reviewer.

**P 16 L 23 –>** *surprisingly* **- considering earlier studies on the freezing of saline solutions I would not rate this as surprising.**

We have removed the word surprising.

[revised manuscript text omitted]

---

## Referee Report (RR1)

**Re-Review of tc-2014-88**

This is a short re-review of the Cryosphere Discussions resubmitted manuscript tc-2015-197 *Direct visualization of solute locations in laboratory ice samples* by Theodore Hullar and Cort Anastasio.

The authors have properly adressed two major issues mentioned in my review. First, the authors have considerably improved the referencing to the published background on ice observations and solute redistribution during freezing and discussed their results within this background. Second, they have properly discussed the limitations of the present study and pointed out the need of more observations and systematic studies of frozen solutions by X-ray computed tomography. I like to congratulate the authors to their work and article that I regard as suitable for publication in The Cryosphere in the present form.